# KDM6B promotes activation of the oncogenic CDK4/6-pRB-E2F pathway by maintaining enhancer activity in MYCN-amplified neuroblastoma

Alexandra D'Oto[1,4], Jie Fang[1,4], Hongjian Jin[2], Beisi Xu [2], Shivendra Singh[1], Anoushka Mullasseril[1], Victoria Jones[1], Ahmed Abu-Zaid[1], Xinyu von Buttlar[1], Bailey Cooke[1], Dongli Hu[1], Jason Shohet[3], Andrew J. Murphy [1], Andrew M. Davidoff [1✉] & Jun Yang [1✉]

The H3K27me2/me3 histone demethylase KDM6B is essential to neuroblastoma cell survival. However, the mechanism of KDM6B action remains poorly defined. We demonstrate that inhibition of KDM6B activity 1) reduces the chromatin accessibility of E2F target genes and *MYCN, 2)* selectively leads to an increase of H3K27me3 but a decrease of the enhancer mark H3K4me1 at the CTCF and BORIS binding sites, which may, consequently, disrupt the long-range chromatin interaction of *MYCN* and E2F target genes, and 3) phenocopies the transcriptome induced by the specific CDK4/6 inhibitor palbociclib. Overexpression of CDK4/6 or *Rb1* knockout confers neuroblastoma cell resistance to both palbociclib and the KDM6 inhibitor GSK-J4. These data indicate that KDM6B promotes an oncogenic CDK4/6-pRB-E2F pathway in neuroblastoma cells via H3K27me3-dependent enhancer-promoter interactions, providing a rationale to target KDM6B for high-risk neuroblastoma.

[1] Department of Surgery, St. Jude Children's Research Hospital, 262 Danny Thomas Place, Memphis, TN 38105, USA. [2] Center for Applied Bioinformatics, St. Jude Children's Research Hospital, 262 Danny Thomas Place, Memphis, TN 38105, USA. [3] Department of Pediatrics, University of Massachusetts Medical School, 55 Lake Avenue North, Worcester, MA 01655, USA. [4]These authors contributed equally: Alexandra D'Oto, Jie Fang ✉email: Jun.Yang2@stjude.org; Andrew.Davidoff@stjude.org

KDM6 is a Jumonji domain-containing H3K27me3/me2 demethylase subfamily[1–4] that antagonizes the activity of polycomb repressive complex 2 (PRC2)[5], the methyl-transferase of H3K27. KDM6 consists of three members, KDM6A, KDM6B and UTY, although UTY has low lysine demethylase activity[6]. While KDM6A is believed to be a tumor suppressor and is mutated in many different types of cancer[7,8], the function of KDM6B in cancer remains poorly defined. Early studies have shown that KDM6B contributes to activation of the *Ink4a/Arf* locus in fibroblasts in response to oncogenic stress[9,10], and nuclear p53 protein stabilization in glioblastoma cells[11]. Loss of KDM6B enhances aggressiveness of pancreatic cancer cells[12]. These data suggest KDM6B might have a tumor suppressive function. However, KDM6B is overexpressed in various cancers[13], indicating that KDM6B may play different roles depending on the cellular context. KDM6B appears to be essential for the initiation and maintenance of NOTCH-driven acute T-cell lymphoblastic leukemia[14], regulates multiple myeloma cell growth and survival by modulating the MAPK pathway[15], drives glioblastoma stem cell plasticity and drug tolerance by chromatin remodeling[16], and promotes migration, invasion, and stem cell-like behaviors in hepatocellular carcinoma[17]. KDM6B is also involved in chemotherapy resistance[18].

Neuroblastoma, the most common extra-cranial solid tumor in children, arises as a result of blocked differentiation of neural crest precursors (NCCs) during development of the sympathetic nervous system[19,20]. This aggressive malignancy accounts for 15% of cancer-related deaths in children[21]. Although outcomes for children with low- or intermediate-risk disease are excellent, fewer than 50% of children with high-risk disease survive despite aggressive multimodal therapy. In patients with high-risk disease, one key biological feature associated with poor prognosis is amplification of the *MYCN* oncogene (MNA)[22]. As a master lineage transcription factor that drives tumorigenesis, the functions of MYCN have been associated with targetable epigenetic modifiers[23–25]. Our previous study demonstrated that the histone demethylase KDM4B regulates MYC activity and promotes tumor growth and maintenance of neuroblastoma[26]. More recently, we identified a KDM4B inhibitor ciclopirox[27], that inhibits tumor growth and promotes differentiation[27]. EZH2, the essential catalytic unit of the PRC2 complex, is a MYCN target that represses neuronal differentiation in a H3K27me3-dependent manner, leading to inactivation of a tumor suppressive program in neuroblastoma[28].

In this study, we show that KDM6B is highly expressed in neuroblastoma and its genomic locus is broadly marked with transcriptionally active histone modifications and transcription factor binding. Using chemical genetic and epigenetic approaches, we show that KDM6B is involved in regulation of the key oncogenic CDK4/6-pRB-E2F pathway and expression of both *MYCN* and *C-MYC* proto-oncogenes. Furthermore, the E2F transcriptome serves as a biological marker of KDM6 inhibitor sensitivity. Pharmacological inhibition of KDM6B activity alters the chromatin accessibility of E2F target genes and *MYCN*, induces the redistribution of H3K27me3 and the enhancer mark H3K4me1. We propose that this may disrupt the long-range chromatin interaction of the E2F transcriptome within the same topologically associated domains (TAD). At the level of transcriptome, KDM6B inhibition mimics the CDK4/6 inhibitor palbociclib. Cancer cells resistant to CDK4/6 inhibitor also are resistant to KDM6B inhibition. The gene signature targeted by KDM6 inhibition was associated with poor patient survival. Thus, our studies reveal that KDM6B regulates the oncogenic CDK4/6-pRB-E2F pathway in MYCN-amplified neuroblastoma, revealing a mechanism of regulation of the E2F transcriptome by an epigenetic modulator.

## Results

### KDM6B is highly expressed in human neuroblastoma and is associated with a transcriptionally active epigenetic landscape.

We have previously shown that *KDM6B*, but not *KDM6A* or *UTY*, is essential for neuroblastoma cell survival[27]. To more clearly define the role of KDM6B in neuroblastoma, we first compared expression of KDM6 family genes in normal trunk neural crest-derived tissues, neuroblastoma being a neural crest-derived cancer of the peripheral nervous system[29], with expression in tumors in four different neuroblastoma patient cohorts[30–32]. While expression of *KDM6A* and *UTY* did not show consistent differences in tumors as compared to normal tissue of neural crest origin, *KDM6B* expression was significantly elevated across all four neuroblastoma cohorts ($p < 0.001$) (Fig. 1a–c). We also found that levels of *KDM6B* expression, but not *KDM6A* or *UTY*, in neuroblastoma cell lines were among the highest, across 40 different cancer lineages (Supplementary Fig. 1a). Analysis of the data from 15 subtypes of pediatric cancers (PCGP project performed at St. Jude) also showed *KDM6B* expression was among the highest in neuroblastoma (Supplementary Fig. 1b). We then compared the RNA-seq transcript counts of *KDM6B*, *KDM6A*, and *UTY* in three large neuroblastoma cohorts and found that expression of *KDM6B* was significantly higher than that of *KDM6A* or *UTY* (Fig. 1d–f).

Gene transcription status is usually associated with a distinct epigenetic landscape. For example, high levels of H3K4me1, H3K4me3, and H3K27Ac are marks of active gene transcription while high levels of H3K27me3 and H3K9me3 are marks of repressive gene transcription[33]. To determine whether high expression of *KDM6B* in neuroblastoma is associated with active epigenetic modifications, we investigated the epigenetic landscapes at the genomic loci of *KDM6B*, *KDM6A*, and *UTY* in seven primary human neuroblastoma tissues sequenced at St. Jude, four with MNA, and three without MNA (Fig. 1g and Supplementary Fig. 2). Regardless of the *MYCN* status, the genomic locus of *KDM6B* across 22-kb in length in all 7 human neuroblastomas is occupied by RNA polymerase II, which drives gene transcription, in accordance with enrichment of active gene transcription marks H3K4me1, H3K4me2, H3K4me3, H3K27Ac, H3K9-K14Ac, and H3K36me3. However, no transcriptionally repressive H3K27me3 or H3K9me3 marks were occupied at the *KDM6B* locus. The broad occupation of H3K4me1 and H3K27Ac across the whole locus of *KDM6B* indicates that a super-enhancer[34,35] may regulate the expression of *KDM6B* in neuroblastoma. In contrast, the epigenetic landscapes at the *KDM6A* and *UTY* loci appeared to lack transcriptionally active gene transcription marks (Fig. 1g and Supplementary Fig. 2). The promoter region and gene body of *KDM6A* were occupied by the transcriptionally repressive marks H3K27me3 and H3K9me3, respectively. In summary, *KDM6B* is highly expressed in human neuroblastoma and its genomic locus is epigenetically modified for active gene transcription, which is distinct from its paralogs *KDM6A* and *UTY*.

### KDM6B regulates MYC expression.

We further validated the importance of KDM6B in neuroblastoma by assessing colony formation after knocking down *KDM6B* with four different siRNAs in BE2C cells (with MNA) and SK-N-AS cells (with high C-MYC activity). KDM6B depletion greatly reduced colony formation in both neuroblastoma cell lines, and was correlated with the KDM6B knockdown efficiency (Fig. 1h, i), consistent with our previous study[27]. In support of our data, neuroblastoma is the third most sensitive among 25 different cancer lineages that are sensitive to knockdown of KDM6B but not KDM6A (Supplementary Fig. 3a, b), just below prostate cancer and myeloma, in a

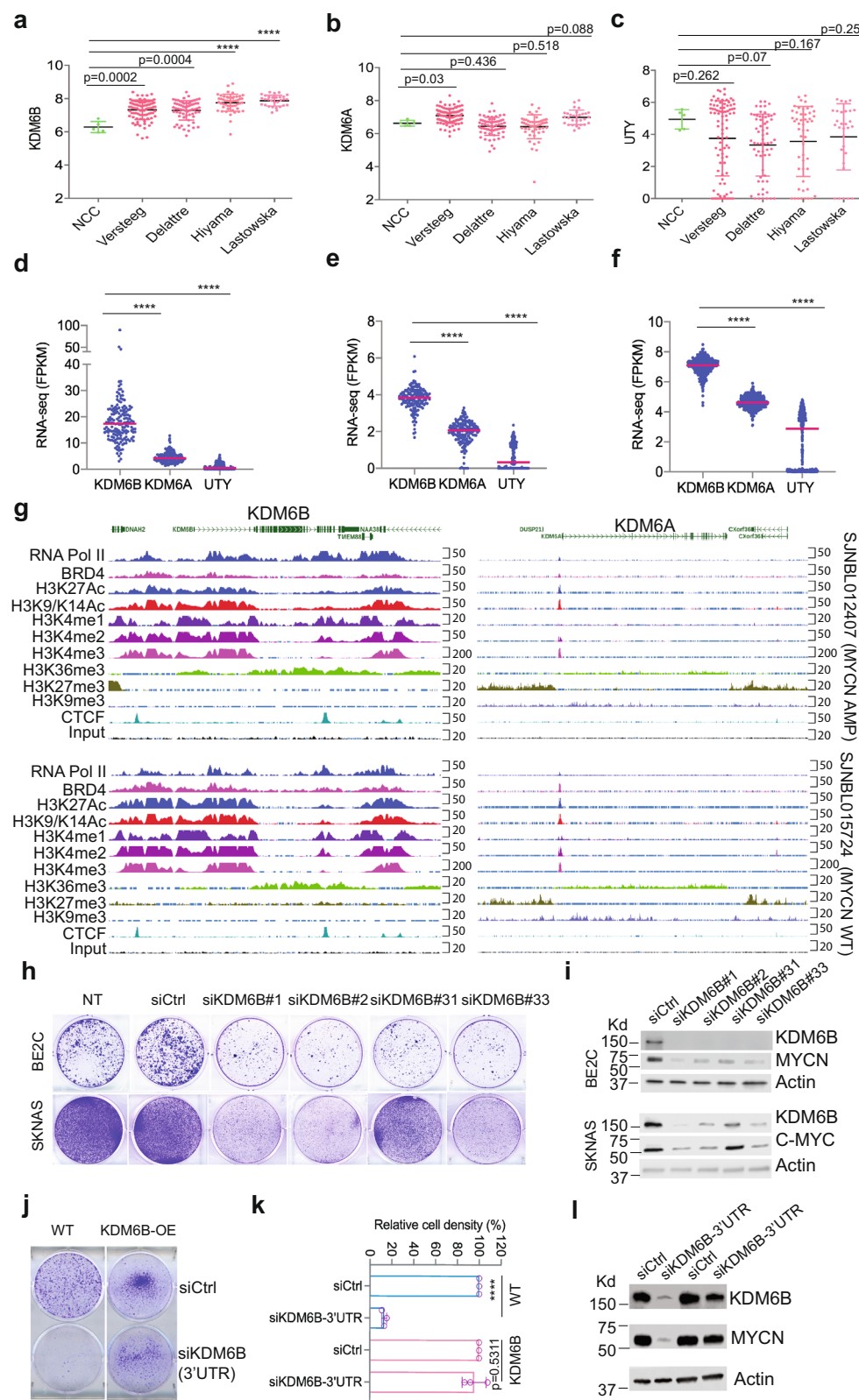

genome-wide shRNA screen (20 shRNAs per gene) in 709 cancer cell lines (DepMap dataset). Notably, depletion of KDM6B led to MYCN and C-MYC reduction in neuroblastoma cells (Fig. 1i and Supplementary Fig. 3c–f). Importantly, introduction of retroviral based *KDM6B* cDNA without its endogenous 3′UTR rescued the colony formation and MYCN expression in BE2C cells when endogenous KDM6B was depleted using a siRNA designed to

target its 3′UTR (Fig. 1j–l), demonstrating the phenotype induced by KDM6B depletion is not due to an off-target effect. Conversely, overexpression of KDM6B in BE2C cells reduced H3K27me3, increased MYCN expression and promoted cell proliferation (Supplementary Fig. 3g, h). In addition, we found that KDM6B also regulated MYC expression in other cancer lineages such as lung cancer, osteosarcoma, and colorectal cancer

**Fig. 1 KDM6B is marked with a highly active epigenetic landscape and highly expressed in neuroblastoma and regulates MYC expression. a–c** The expression of *KDM6B*, *KDM6A*, and *UTY* in normal human trunk neural crest (GSE14340) and four different neuroblastoma cohorts (Versteeg GSE16476, Delattre GSE14880, Hiyama GSE16237, Lastowska GSE13136). y-Axis represents the normalized log2 expression value. n = 5 for NCC, n = 88 for Versteeg, n = 64 for Delattre, n = 51 for Hiyama, n = 30 for Lastowska. Data are represented as mean ± SD. ****p < 0.0001, two-tailed, unpaired *t* test. **d–f** The expression of *KDM6B*, *KDM6A*, and *UTY* in three different neuroblastoma RNA-seq cohorts. The RNA-seq data of St Jude (**d**) was downloaded from https://pecan.stjude.cloud. The RNA-seq data of TARGET (**e**) and SEQC (**f**) datasets were downloaded from R2 (https://hgserver1.amc.nl/cgi-bin/r2/main.cgi). y-Axis represents the Fragment Per Kilobase of transcript per Million (FPKM) mapped reads. n = 160 for St Jude dataset, n = 161 for TARGET dataset, n = 498 for SEQC dataset. Data are represented as mean ± SD. ****p < 0.0001, two-tailed, unpaired *t* test. **g** The epigenetic landscapes consisting of histone marks and transcription factor binding distinguish *KDM6B* from *KDM6A* in primary neuroblastoma tissues with *MYCN* amplification or without *MYCN* amplification (https://pecan.stjude.cloud/proteinpaint/study/mycn_nbl_2018). **h** Crystal violet staining of colonies after BE2C and SK-N-AS cells were transfected with four different siRNA oligos to knockdown *KDM6B* for 7 days. siCtrl = siRNA control oligo, NT = no treatment. **i** Western blot analysis with indicated antibodies to assess MYCN or C-MYC expression after 3-day transfection of 4 different siRNA to knockdown KDM6B in BE2C and SK-N-AS. The blots are representative of three independent experiments. **j** BE2C and MSCV-KDM6B overexpressing (KDM6B-OE) BE2C cells were transfected with siRNA control (siCtrl) and siRNA targeting the endogenous 3′ untranslated region (3′UTR) of KDM6B (siKDM6B-3′UTR). Four days later, cells were stained with crystal violet. **k** Quantification of cell density of each group (n = 3) using imageJ. Data are represented as mean ± SD. Shown are individual biological replicates. ****p < 0.0001, two-tailed, unpaired *t* test. **l** BE2C and MSCV-KDM6B overexpressing (KDM6B-OE) BE2C cells were transfected with siRNA control (siCtrl) and siRNA siKDM6B-3′UTR. Three days later, cells were subject to immunoblotting with indicated antibodies. The blots are representative of three independent experiments. Source data are provided as a "Source data" file.

| Table 1 Transcription factor binding motif enrichment for KDM6B target genes (top 15). | | | |
|---|---|---|---|
| **Gene sets** | **NES** | **NOM p-val** | **FDR q-val** |
| 1 KRCTCNNNNMANAGC_UNKNOWN | 2.76 | 0 | 0 |
| 2 SGCGSSAAA_E2F1DP2_01 | 2.17 | 0 | 0 |
| 3 TTTNNANAGCYR_UNKNOWN | 2.14 | 0 | 0 |
| 4 E2F4DP1_01 | 2.1 | 0 | 0 |
| 5 E2F_03 | 2.06 | 0 | 0 |
| 6 E2F1_Q6 | 2.03 | 0 | 0 |
| 7 E2F_02 | 2.03 | 0 | 0 |
| 8 E2F_Q6 | 2.02 | 0 | 0 |
| 9 E2F_Q3 | 2.01 | 0 | 0 |
| 10 E2F4DP2_01 | 2 | 0 | 0 |
| 11 E2F1DP2_01 | 2 | 0 | 0 |
| 12 E2F1DP1_01 | 1.99 | 0 | 0 |
| 13 E2F_Q4 | 1.99 | 0 | 0 |
| 14 E2F_Q6_01 | 1.92 | 0 | 0 |
| 15 E2F1_Q6_01 | 1.9 | 0 | 0 |

Gene set enrichment scores analyzed by GSEA program. *NES* normalized enrichment score, *NOM p-val* nominal *p* value, *FDR* false discovery rate. *p* Value calculated by one-sided Fisher's exact test. The FDR is calculated by comparing the distribution of normalized enrichment scores from many different genesets.

cells and that this was independent of p53 status (Supplementary Fig. 3i–l). These data indicate that KDM6B is needed for MYC expression in cancer cells.

**KDM6B regulates a pRB-E2F transcription program in neuroblastoma.** To investigate the molecular function of KDM6B in neuroblastoma, we performed RNA-seq followed by gene set enrichment analysis (GSEA) after knockdown of KDM6B in MYCN-amplified (BE2C and KELLY) and MYCN-non-amplified cells (SK-N-AS, SK-N-FI). Strikingly, GSEA results showed that the top hits of KDM6B downstream targets were enriched with E2F gene sets, *Rb1* knockout signatures (*Rb1* encodes pRB, is a negative regulator of E2F activity[36]) and other cell cycle gene signatures (Fig. 2a–c and Supplementary Fig. 4a, b), in the two MYCN amplified cell lines. Motif analysis of the KDM6B target genes also revealed that they were mainly enriched with E2F-binding motifs (Table 1). These data indicate that KDM6B regulates the transcriptome of the E2F pathway in MYCN-amplified

neuroblastoma cells. The E2F transcription factors are critical regulators of cell cycle progression by directly regulating more than 100 genes including those encoding cyclins[37]. Early studies showed that E2F also binds to promoters of *C-MYC* and *MYCN* to regulate their expression[38,39]. GSEA data revealed that MYC targets were also significantly downregulated by KDM6B depletion (Fig. 2a). Interestingly, the top signatures upregulated by KDM6B knockdown in SK-N-FI were enriched with E2F gene sets (Supplementary Fig. 4c, d), although C-MYC expression was reduced (Supplementary Fig. 3f). However, the E2F gene signatures were less affected in SK-N-AS cells than BE2C, KELLY, and SK-N-FI although one E2F3 gene set was downregulated by KDM6B knockdown (Supplementary Fig. 4e, f).

We further investigated KDM6B function using pharmacologic inhibition. GSK-J1 is the only selective KDM6 inhibitor available[40], but, unfortunately, it is unable to penetrate cells. To make GSK-J1 cell-permeable, GSK-J1 was modified by adding an ester (new reagent was named GSK-J4)[40]. Thus, GSK-J4 is not itself a chemical tool for direct KDM inhibition, but was designed specifically to enable efficient intracellular delivery of GSK-J1 into cells. The intracellular conversion of the ester pro-drug is complete within 15 min after which levels of intracellular GSK-J4 are negligible[41]. Once entering cells, GSK-J4 converts back to GSK-J1, consequently inhibiting KDM6 (Supplementary Fig. 5a)[40]. To validate this and profile the selectivity of GSK-J1/ GSK-J4, we used an AlphaLISA approach, a method that tests the in vitro inhibitory activity of GSK-J1 and GSK-J4 against purified KDMs (KDM2A, KDM3A, KDM4A, KDM5A, KDM6A, and KDM6B). We confirmed that GSK-J1 was selectively effective against KDM6A and KDM6B, had some effect on KDM5A but showed no activity against KDM2A, KDM3A, and KDM4A (Supplementary Fig. 5b), a result consistent with published GSK data[40]. However, GSK-J4 showed no in vitro activity against all tested KDMs (Supplementary Fig. 5c). While GSK-J4 could have off-target effects when administered at high concentrations, the concentration we used in cells (2.5 μM) only induced the upregulation of H3K27me3, but not H3K9me3, H3K36me3, and H3K4me3 (Supplementary Fig. 5d), three other major histone methylation sites of H3, indicating that GSK-J4 induced its cellular effect by selectively targeting KDM6 but not other KDMs. In addition, a selective KDM5 inhibitor, KDM5-C70[42], showed no effect on neuroblastoma cell proliferation even when administered at very high concentrations (40 μM) (Supplementary Fig. 5e, f).

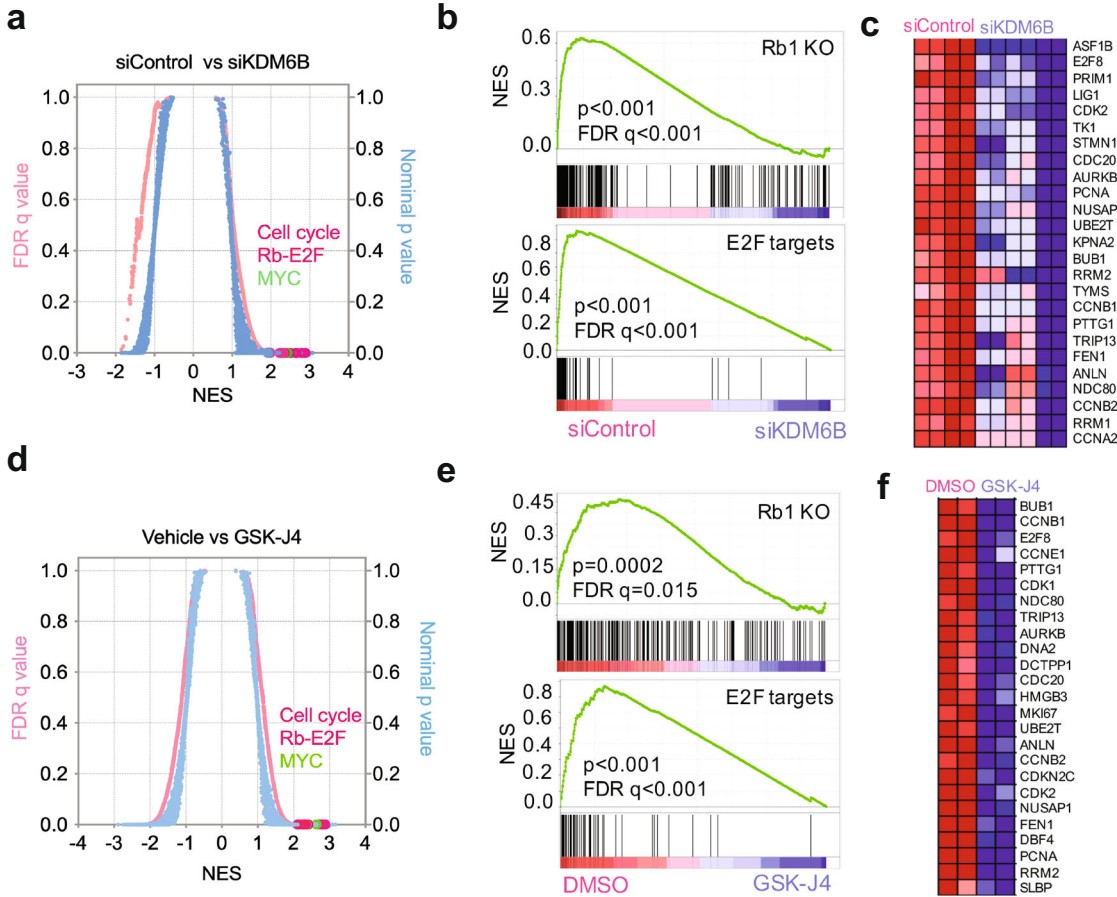

**Fig. 2 KDM6B predominantly regulates the pRB-E2F pathway. a** Quantitative comparison of all chemical and genetic perturbation gene sets ($n = 3403$) from the MSigDB by gene set enrichment analysis (GSEA)[92] for increased (left) and reduced (right) expression of global genes caused by KDM6B knockdown. Data are presented as a scatterplot of normalized $p$ value (right $y$-axis)/false discovery $q$ value (left $y$-axis) vs. normalized enrichment score (NES) ($x$-axis) for each evaluated gene set. The gene sets circled in red color indicate cell cycle, pRB-E2F, and MYC pathway gene sets. $p$ Value calculated by one-sided Fisher's exact test. The FDR is calculated by comparing the distribution of normalized enrichment scores from many different genesets. **b** Two examples of GSEA from **a** show that genes downregulated by depletion of *KDM6B* are enriched with *Rb1* knockout and E2F targets. $p$ Value calculated by one-sided Fisher's exact test. The FDR is calculated by comparing the distribution of normalized enrichment scores from many different genesets. **c** Heatmap shows the gene list from the E2F targets (**b**). **d–f** Similar analysis for GSK-J4 treatment as shown in (**a–c**). Source data are provided as a "Source data" file.

Treatment of neuroblastoma cells and normal human fibroblasts (HS68) with GSK-J4, in a colony formation assay, showed that GSK-J4 inhibited colony formation in a concentration-dependent manner, with a significant effect observed in the majority of cell lines at concentrations of 1.0–2.0 µM (Supplementary Fig. 6a). However, HS68 cells were unaffected by GSK-J4. Consistent with these results, another recent study showed that GSK-J4 has a potent effect on neuroblastoma cell survival[43]. To define the molecular mechanism of GSK-J4 action on neuroblastoma cells, we performed global gene expression profiling for GSEA analysis after GSK-J4 treatment of BE2C cells. GSK-J4 induced a very similar transcriptome profile as depletion of KDM6B and the gene sets most significantly downregulated by this inhibitor were enriched with cell cycle, *Rb1* knockout and E2F pathways (Fig. 2d–f). Transcription factor motif binding analysis showed that genes downregulated by GSK-J4 were greatly enriched with E2F binding (Table 2). Again, MYCN and C-MYC expression were also downregulated by GSK-J4 treatment (Supplementary Fig. 6b–d), consistent with KDM6B depletion (Supplementary Fig. 3). Re-analysis of an independent RNA-seq data set[43] also showed that KDM6 inhibition led to downregulation of gene sets enriched with *Rb1* knockout and DREAM complex (dimerization partner, RB-like, E2F, and multi-vulval class B) in neuroblastoma cells (Supplementary Fig. 7). Thus, GSK-J4 phenocopied KDM6B depletion in neuroblastoma cells and, therefore, its therapeutic effect may be mediated through inhibition of KDM6B. Taken together, genetic and pharmacologic inhibition of KDM6B predominantly leads to downregulation of the pRB-E2F transcriptional program, particularly in MYCN-amplified neuroblastoma cells.

**Pharmacogenetics reveals that the E2F transcriptome is associated with sensitivity to KDM6B inhibition.** While the presence of KDM6B is a prerequisite for GSK-J4 activity in neuroblastoma cells[43], expression levels of KDM6B do not correlate with GSK-J4 sensitivity[43]. A biomarker that predicts therapy response is critical for patient stratification. Since KDM6B inhibition appears to affect its antitumor activity through downregulation of the E2F transcriptome, we hypothesized that high levels of the E2F transcriptome may correlate with the response of cancer cells to KDM6B inhibition. The gene signatures induced by chemical (i.e., small-molecule) or genetic perturbations (i.e., siRNA knockdown) can be used to connect unknown mechanisms of action (MoA) of chemical probes.

**Table 2 Transcription factor binding motif enrichment for GSK-J4 target genes (top 15).**

| | Gene sets | NES | NOM p-val | FDR q-val |
|---|---|---|---|---|
| 1 | KRCTCNNNNMANAGC_UNKNOWN | 2.43 | 0 | 0 |
| 2 | SGCGSSAAA_V$E2F1DP2_01 | 1.48 | 0.003 | 0.253 |
| 3 | V$E2F_01 | 1.42 | 0.037 | 0.31 |
| 4 | V$E2F_Q3 | 1.39 | 0.012 | 0.325 |
| 5 | TTTNNANAGCYR_UNKNOWN | 1.39 | 0.023 | 0.262 |
| 6 | V$E2F_Q6 | 1.36 | 0.012 | 0.274 |
| 7 | V$E2F1_Q3 | 1.33 | 0.015 | 0.308 |
| 8 | V$E2F4DP1_01 | 1.32 | 0.018 | 0.301 |
| 9 | V$E2F1_Q6 | 1.32 | 0.02 | 0.269 |
| 10 | V$E2F_Q4 | 1.31 | 0.013 | 0.273 |
| 11 | RGTTAMWNATT_V$HNF1_01 | 1.31 | 0.071 | 0.25 |
| 12 | V$E2F1DP2_01 | 1.29 | 0.027 | 0.268 |
| 13 | V$E2F_02 | 1.28 | 0.028 | 0.266 |
| 14 | V$E2F1_Q6_01 | 1.28 | 0.042 | 0.255 |
| 15 | V$E2F1DP1_01 | 1.28 | 0.037 | 0.246 |

Gene set enrichment scores analyzed by GSEA program. *NES* normalized enrichment score, *NOM p-val* nominal p value, *FDR* false discovery rate. p Value calculated by one-sided Fisher's exact test. The FDR is calculated by comparing the distribution of normalized enrichment scores from many different genesets.

Differential gene expression has been correlated with patterns of small-molecule sensitivity across many cell lines to illuminate the actions of compounds whose MoA are unknown[44,45]. The Cancer Therapeutic Response Portal correlated the sensitivity patterns of 481 compounds including GSK-J4 with 19,000 basal transcript levels across 823 different human cancer cell lines and identified selective outlier transcripts[44,45], which allowed us to interrogate whether the KDM6B inhibitor activity was correlated with specific transcripts. The E2F genes such as *E2F8* were the ones most significantly affected by KDM6B inhibition (Fig. 2). E2F8 is an atypical E2F and downstream target of E2F1–3[46]. We chose *E2F8* as a representative gene to correlate E2F pathway activity with drug effect. Strikingly, GSK-J4 ranked highest among the top compounds whose cellular activity was correlated with *E2F8* expression (Pearson correlation $R = -0.528$, $z$-score $= -4.56$, Fig. 3a). We arbitrarily chose 13 cell lines with high *E2F8* expression as group A, and 8 cell lines with low *E2F8* expression as group B, which had high sensitivity and low sensitivity to GSK-J4, respectively (Fig. 3b and Supplementary Table 1). We then used GSEA to infer the pathways that differed in the two groups. We found that the pRB-E2F gene sets were the most significant hits in group A cell lines (Fig. 3c, d). The genes highly expressed in group A cells were profoundly enriched with E2F transcription factor binding motifs (Supplementary Table 2), and most of the top 50 genes such as *E2F1* and *E2F8* are involved in regulation of the cell cycle (Supplementary Fig. 8a). MYC signatures were also enriched in group A cells (Supplementary Fig. 8b–d). We also selectively investigated several other E2F genes including *ASF1B*, *CDK2* and *E2F1*, all of which were positively correlated with GSK-J4 sensitivity (Supplementary Fig. 8e–g). Therefore, cancer cells with high E2F activity appear to be more sensitive to KDM6B inhibition.

To specifically investigate the conclusion that high E2F activity serves as a biomarker for sensitivity to KDM6B inhibition in neuroblastoma, we categorized neuroblastoma cell lines into two groups according to their IC50 to GSK-J4[43]. GSEA analysis of RNA-seq data of these cell lines showed a very similar transcriptome pattern to KDM6B knockdown or GSK-J4 inhibition for those with lower IC50 to GSK-J4 (more sensitive), which were enriched with E2F and MYC gene signatures (Fig. 3e). For example, SK-N-FI cells with high E2F transcriptome

expression were more sensitive to GSK-J4 than SK-N-SH cells (Fig. 3f), which had lower expression of the E2F transcriptome. The genes highly expressed in neuroblastoma cells that were more sensitive to GSK-J4 were predominantly enriched with E2F transcription factor binding motifs (Supplementary Table 3). Taken together, these data demonstrate that a high E2F gene signature is correlated with GSK-J4 sensitivity, and thus it may serve as a biomarker for effectiveness of KDM6B inhibitors in cancer treatment in the future. These data also further support that KDM6B regulates the E2F transcriptome.

**Inhibition of KDM6B activity reduces the chromatin accessibility of E2F target genes and MYCN.** To understand how KDM6B regulates the E2F pathway, we performed Assay for Transposase-Accessible Chromatin with high-throughput sequencing (ATAC-seq) after 24 and 48 h of GSK-J4 treatment, for mapping genome-wide chromatin accessibility[47]. The 24 h GSK-J4 treatment resulted in upregulation of 8386 nucleosome-free, open chromatin regions (log2 fold change ≥ 0.5, $p < 0.05$, Fig. 4a), 46.9% of which were located at the annotated promoter regions (within 2-kb from transcription start site, TSS) and 39% of which were located at the annotated enhancer regions (2–50 kb distance from TSS). The rest (14.1%) were located at more distal regions. However, 9574 nucleosome free, open chromatin regions were downregulated by GSK-J4 treatment (Fig. 4a), only 18.5% of which were located at the promoter regions and 61.8% of which were located at enhancer regions, while 19.7% were located at more distal regions. The 48 h GSK-J4 treatment further enhanced its effect on DNA accessibility, resulting in an increase of 13,546 and decrease of 11027 peaks, respectively (Fig. 4a). The proportions of the annotated regions tended to be similar to that of 24 h treatment. Among the regions with upregulated chromatin accessibility, 57.6% were located at promoter regions, 29.5% were located at enhancer regions, and 12.9% were located at more distal regions (Fig. 4a). Among the regions with downregulated chromatin accessibility, 17.9% were located at promoter regions and 57.8% were located at enhancer regions, while 24.3% were located at more distal regions (Fig. 4a). We further assessed whether altered chromatin accessibility was associated with changes in gene expression. Indeed, the genes annotated with differential accessibility regions (DARs) at promoters or enhancers were significantly enriched in differentially expressed genes by GSK-J4 treatment (Supplementary Fig. 9). For example, DNA accessibility at the promoter region of the oncogene *AKT1* was reduced by GSK-J4, and one additional peak at a potential enhancer region near the long noncoding RNA *LINC00638* was also decreased (Fig. 4b). The reduction of chromatin accessibility was consistent with the reduced expression of *AKT1* transcripts. AKT1 is a key molecule in the PI3K-mTOR pathway[48], and E2F upregulates AKT activity through a transcription-dependent mechanism[49]. *E2F8*, whose expression was downregulated by KDM6B knockdown and GSK-J4 treatment, did not show significant changes in DNA accessibility within its promoter region. However, several peaks at the distal region of E2F8 which extended to the adjacent *NAV2* locus were altered by GSK-J4 (Fig. 4c). Notably, our RNA-seq analysis showed that the *NAV2* locus is not expressed in BE2C cells (Fig. 4c). These data suggest that *E2F8* expression might be regulated by a distal enhancer that is silenced by KDM6B inhibition. Similarly, we did not observe significant changes at the promoter regions of *MYCN*, and cell cycle genes (*CCNE1*, *BUB1*, and *BIRC5*); however, the DNA accessibility at the distal regions of these genes were reduced (Supplementary Fig. 10a–d). The downregulation of gene expression by GSK-J4 was not always correlated with reduction of DNA accessibility. For example, chromatin accessibility at the

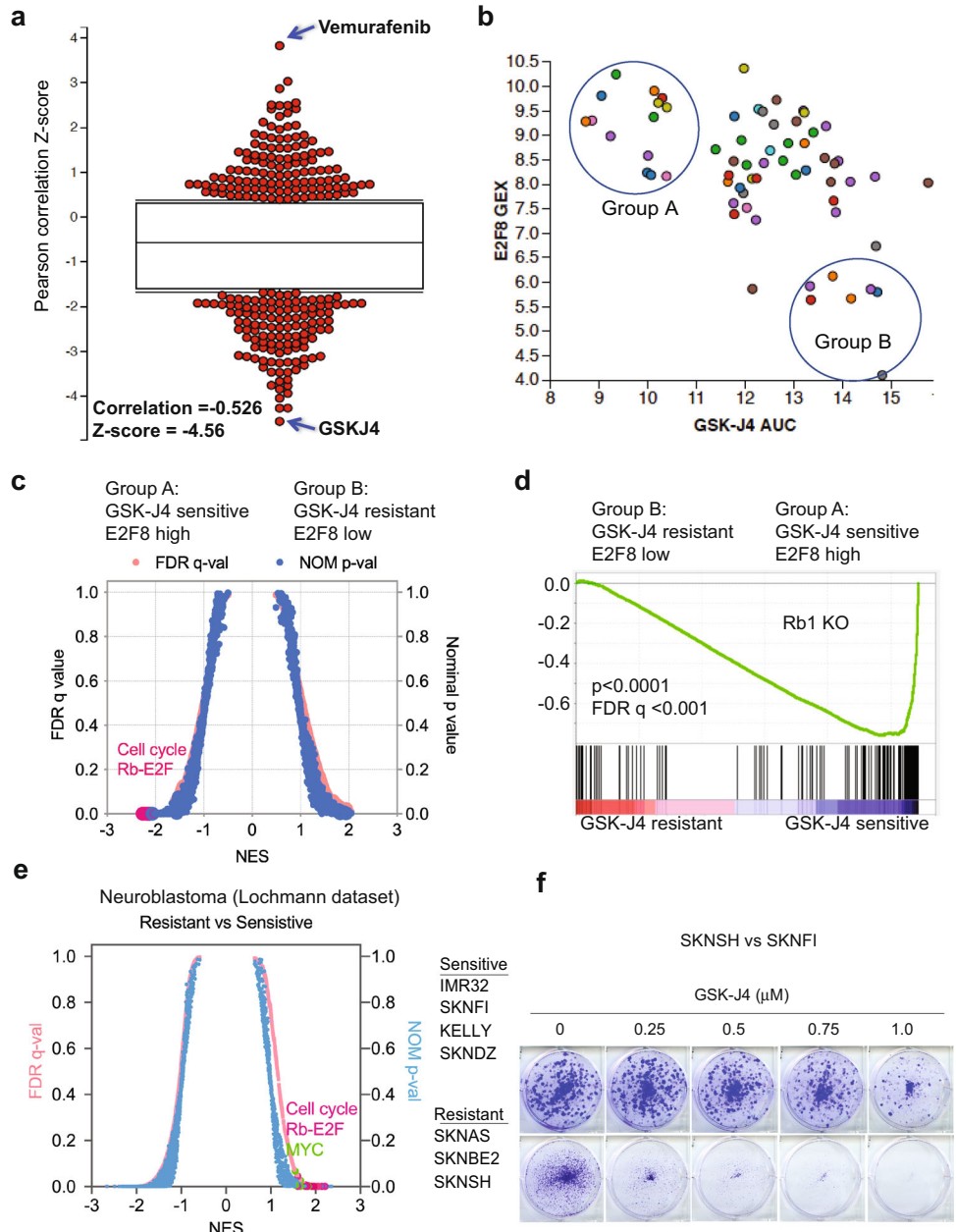

**Fig. 3 Chemogenetic data show E2F gene signature is correlated with sensitivity of GSK-J4. a** The Pearson correlation Z-score for each tested compound with *E2F8* expression, which was extracted from The Cancer Therapeutics Response Portal (CTRP)[44,45]. The correlation of GSK-J4 and *E2F8* ranked on the top with $R = -0.626$, $Z$-score $= -4.56$. $n = 481$ for compounds. The box plot indicates Interquantile Multiplier $= 0$. **b** The plot of *E2F8* expression and GSK-J4 sensitivity for each tested cell line. The *y*-axis represents the transcript expression levels of *E2F8* in each cell line. The *x*-axis represents the drug response metrics (area under curve, AUC) to GSK-J4. The lower the AUC, the more sensitive to GSK-J4. The circled group A and B populations are arbitrarily chosen as GSK-J4 sensitive vs resistant. **c** Quantitative comparison of all chemical and genetic perturbation gene sets ($n = 3403$) from the MSigDB by gene set enrichment analysis (GSEA) for GSK-J4 sensitive (group A) and resistant (group B). Data are presented as a scatterplot of normalized *p* value/false discovery *q* value vs. normalized enrichment score (NES) for each evaluated gene set. *p* Value calculated by one-sided Fisher's exact test. The FDR is calculated by comparing the distribution of normalized enrichment scores from many different genesets. The red dots highlight cell cycle and Rb-E2F pathway gene sets. **d** GSEA shows that genes highly expressed in GSK-J4 sensitive (group A) cells are enriched with Rb1 knockout. **e** Quantitative comparison of all chemical and genetic perturbation gene sets ($n = 3403$) from the MSigDB by gene set enrichment analysis (GSEA) for GSK-J4 sensitive and resistant neuroblastoma cells, as shown in the right panel of cell lines. Data are presented as a scatterplot of normalized *p* value/false discovery *q* value vs. normalized enrichment score (NES) for each evaluated gene set. *p* Value calculated by one-sided Fisher's exact test. The FDR is calculated by comparing the distribution of normalized enrichment scores from many different genesets. The red and green dots highlight cell cycle and Rb-E2F pathway gene sets and MYC gene sets, respectively. **f** Crystal violet staining of colonies after SK-N-SH and SK-N-FI neuroblastoma cell lines were treated with different concentrations of GSK-J4 for 7 days. Source data are provided as a "Source data" file.

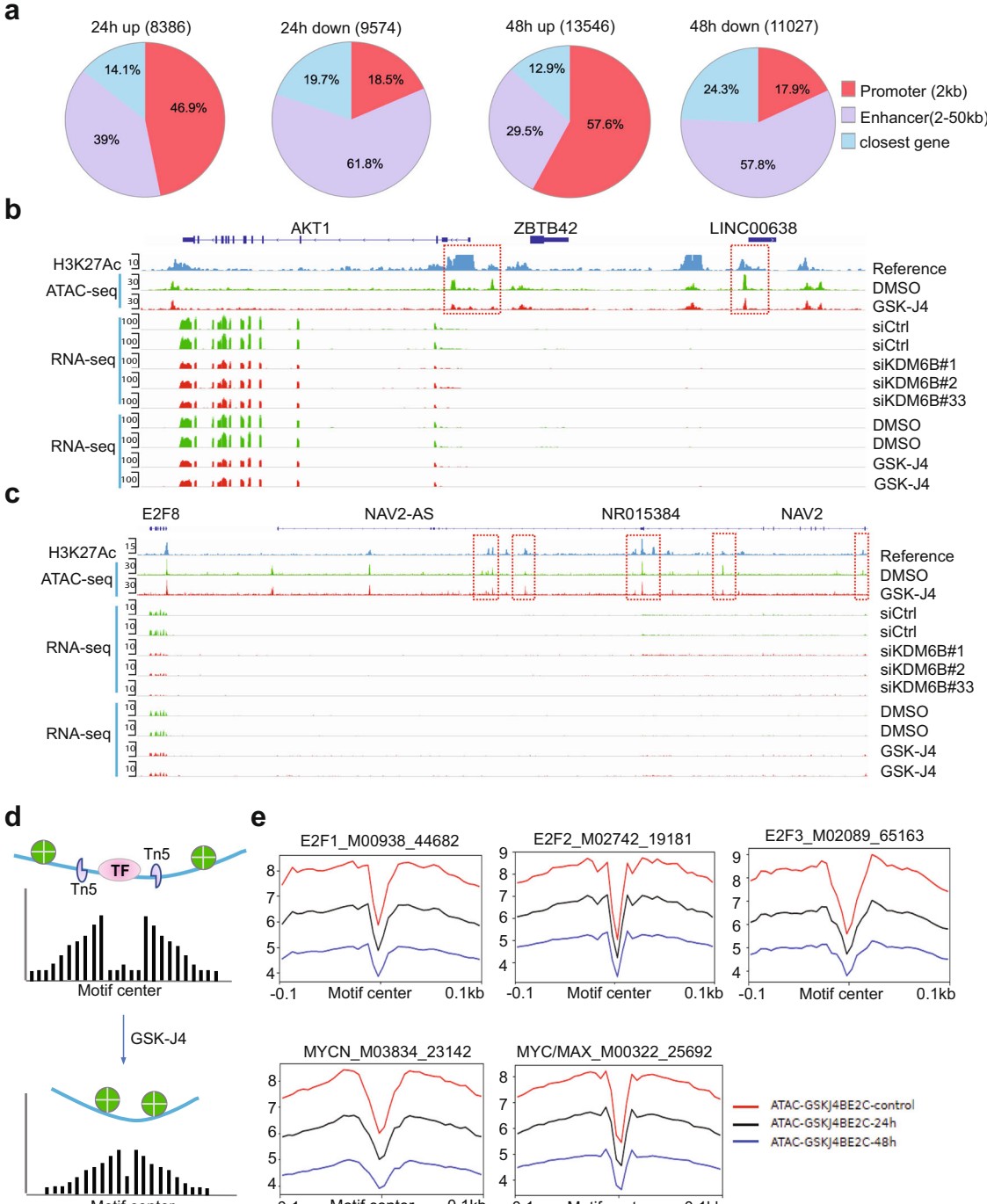

**Fig. 4 KDM6B inhibition represses chromatin accessibility of E2F genes. a** ATAC-seq was performed after BE2C cells were treated with 2.5 μM of GSK-J4 for 24 and 48 h. Summary of peak calling number of ATAC-seq (cut-off $p < 0.05$, log2 fold change $\geq$ 0.5) including the upregulated and downregulated nucleosome free regions, and the annotated locations of the peaks at defined promoter and enhancer regions. $p$ Value obtained by one-sided Empirical Bayes Statistics test. **b** Snapshot of *AKT1* locus using Integrative Genomic Viewer (IGV) for ATAC-seq, H3K27Ac ChIP-seq, and RNA-seq. The ATAC-seq analysis shows the downregulation of two peaks at the 5′ promoter region of *AKT1* and one peak at the non-coding RNA *LINC00638*. The RNA-seq results showed that the AKT1 transcript was downregulated by 48 h of GSK-J4 treatment and KDM6B knockdown. H3K27Ac in BE2C cells was referenced to GSM2113518[109]. **c** Snapshot of *E2F8* locus using Integrative Genomic Viewer (IGV) for ATAC-seq, H3K27Ac ChIP-seq, and RNA-seq. The ATAC-seq shows the downregulation of three peaks at the enhancer region of *E2F8*, next to the *NAV2* gene locus. The RNA-seq results showed that *E2F8* transcript was downregulated by KDM6B inhibition while the adjacent *NAV2* expression is barely detectable. **d** Cartoon indicates the rationale of footprinting analysis. The DNA motifs bound by transcription factors such as E2F1 protect the cut from transposase Tn5, while the adjacent open chromatin gives rise to a high signal of nucleosome free region after ATAC-seq analysis. After GSK-J4 treatment, the open chromatin was repressed and consequently reducing the reads of free DNA. **e** Footprinting plot shows the reduction of open chromatin at predicted binding motifs of E2Fs and MYC after BE2C cells were treated with GSK-J4 for 24 and 48 h. Source data are provided as a "Source data" file.

*AURKB* promoter was increased by GSK-J4 although its transcript expression was reduced (Supplementary Fig. 10e), suggesting that the open chromatin may recruit transcription suppressors to repress its gene transcription. For *CDKN1A*, which encodes p21 to inhibit the CDK2/Cyclin E complex during S phase of the cell cycle, the DNA accessibility at its promoter and enhancer regions was enhanced, consistent with its transcript induction by KDM6B inhibition (Supplementary Fig. 10f).

Transcription factor motif analysis of the ATAC-seq data using genomic footprinting demonstrated that E2F binding motifs had reduced DNA accessibility (Fig. 4d, e). Downregulated open chromatin regions were also observed for MYCN and MYC/MAX binding motifs (Fig. 4e). These data indicate that KDM6B inhibition impacts the chromatin accessibility and transcription of E2F and MYC pathway genes.

**KDM6B inhibition induces an increase of H3K27me3 but a decrease of the enhancer mark H3K4me1 at CTCF- and BORIS-binding sites.** To examine the impact of KDM6B inhibition on epigenetic marks, we performed Cleavage Under Targets & Tagmentation (CUT&Tag) to assess the alterations of H3K27me3, the KDM6B substrate, and H3K4me1, the enhancer mark, after KDM6B knockdown and GSK-J4 treatment of BE2C cells. CUT&Tag is an epigenomic profiling strategy that overcomes shortcomings of ChIP-seq[50]. In CUT&Tag, the H3K27me3 and H3K4me1 were bound in situ by specific antibodies, which were then tethered to a protein A-Tn5 transposase fusion protein. Activation of the transposase efficiently generates fragment libraries with high resolution and exceptionally low background[50]. The CUT&Tag library size for each sample was adjusted by *E. coli* residual genomic DNA as a spike-in scaling factor, which was used to normalize the read counts for the downstream differential analysis. KDM6B inhibition induced a modest increase of global H3K27me3 peaks (Supplementary Data 1 and Supplementary Fig. 11a). We specifically assessed the H3K27me3 in the differentially expressed genes induced by KDM6B inhibition. Among the genes differentially regulated by KDM6B knockdown and GSK-J4 treatment, the patterns of H3K27me3 changes tended to be similar in both groups (Fig. 5a), suggesting that GSK-J4 mainly acts through KDM6B to modulate the H3K27me3 levels. In particular, the H3K27me3 signals at the super-enhancer region of *MYCN* (the genomic region between *MYCN* and *FAM49A*, with a noncoding RNA *GACAT3* between both of them[23,51]) were most significantly upregulated by KDM6B knockdown or GSK-J4 treatment (Fig. 5c and Supplementary Fig. 11a), which is consistent with the downregulation of MYCN by KDM6B inhibition. We further assessed H3K4me1, the enhancer mark, using CUT&Tag after KDM6B knockdown and GSK-J4 treatment (Supplementary Data 2). Similar patterns of H3K4me1 changes were observed in both groups (Fig. 5b). Interestingly, while more peaks of H3K4me1 were increased in genes downregulated by KDM6B inhibition, the super-enhancer region of *MYCN* showed most significant downregulation of H3K4me1 although GSK-J4 also induced an increase of several peaks at this region (Fig. 5c and Supplementary Fig. 11b), suggesting that *MYCN* super-enhancer region is particularly susceptible to perturbation of KDM6B inhibition.

The alterations of H3K27me3 and H3K4me1 by KDM6B inhibition mainly occurred at non-promoter regions (Supplementary Fig. 11c), suggesting that KDM6B has a major impact on distal regulatory elements of transcription. Transcriptional factor motif analyses of H3K27me3 and H3K4me1 peaks revealed that KDM6B inhibition dominantly led to an increase of H3K27me3 but a decrease of H3K4me1 at the CTCF- and BORIS-binding sites (Fig. 5d, e), and no enrichment with MYCN- or E2F-binding

sites. CTCF and BORIS (also named CTCFL) play an important role in chromatin looping and long-range chromatin interactions[52–54]. H3K4me1 is an essential feature of enhancers and is bound by multiple chromatin-associated factors such as BAF complex and Cohesion[55], which facilitate enhancer-promoter looping[56,57]. Since the majority of upregulated H3K27me3 peaks were located at distal regulatory regions, we hypothesized that the H3K27me3 changes at these distal regions repressed transcription of *MYCN* and E2F target genes via alteration of chromatin structure and enhancer/promoter interactions. To investigate this possibility, we cross-referenced the Hi–C chromatin conformation of BE2C cells. Interestingly, *MYCN* is located at the same topologically associated domain (TAD) as the *GACAT3 and FAM49A* loci, marked as a super-enhancer by broad H3K27Ac occupancy, showing dysregulated H3K27me3 and H3K4me1 by KDM6B inhibition (Fig. 5c). A TAD domain is a self-interacting genomic region that regulates gene expression by limiting the enhancer–promoter interaction to each TAD[58,59]. Chromatin interaction was observed across this region (Fig. 5f), supporting the hypothesis that KDM6B inhibition may disrupt the chromatin interaction, leading to reduced transcription of *MYCN*. Similarly, we found the *E2F8* gene locus showed multiple chromatin interactions with adjacent genes within the TAD, particularly to the *NAV2* locus (Supplementary Fig. 11d, e). Taken together, these data suggest that KDM6B inhibition might alter the long-range interactions of *MYCN* and E2F target genes in TAD, repressing the MYCN and E2F regulated transcriptome.

**GSK-J4 reduces the expression of the PRC2 complex.** The homeostasis of H3K27me3 is balanced by PRC2 and KDM6[5]. More H3K27me3 peaks were expected by KDM6 inhibition. However, GSK-J4 treatment did not lead to expected high number loci of H3K27me3 (Supplementary Fig. 11a). To explain this discrepancy, we examined the expression of the PRC2 complex. The key PRC2 components (EZH2, SUZ12, and EED) have been shown to be pRB-E2F targets[60,61]. Consistent with the data that KDM6B inhibition mainly downregulated an pRB-E2F transcriptome, GSK-J4 treatment indeed reduced the expression of the PRC2 complex (Supplementary Fig. 12a), However, the expression of global H3K27me3 was minimally affected, indicating that the expected H3K27me3 upregulation by KDM6 inhibition is counterbalanced by reduced expression of PRC2. A recent study identified a 37-gene signature of EZH2[28], which is repressed by EZH2 and silenced in *MYCN*-amplified high-risk neuroblastoma. GSEA analysis revealed that this 37-gene signature was significantly associated with the GSK-J4 treatment (Supplementary Fig. 12b, c), suggesting that KDM6B inhibition consequently downregulates the functions of PRC2, leading to the induction of tumor-suppressive program repressed by EZH2.

**Inhibition of KDM6B mimics CDK4/6 inhibitor palbociclib.** To further investigate the biological functions of KDM6B, we integrated the KDM6B downstream target genes with information from the Library of Integrated Network-Based Cellular Signatures (LINCS) data. LINCS data are composed of tens of thousands of gene sets, indicative of the transcriptional responses to a large library of chemical compounds[62]. With this approach, we found that genes downregulated by KDM6B knockdown in neuroblastoma cells (Supplementary Table 4) overlapped significantly with the transcriptomes downregulated by palbociclib (Fig. 6a), a specific CDK4/6 inhibitor that acts upstream of the E2F pathway in cell cycle regulation. Similarly, genes downregulated by GSK-J4 (Supplementary Table 5) also significantly overlapped the transcriptomes downregulated by palbociclib

(Fig. 6b). CDK4/6 inhibitor signatures being identified among the most correlated signatures with KDM6B knockdown or GSK-J4 treatment further demonstrated that KDM6B is mainly involved in regulation of the E2F pathway. To corroborate these findings, we treated four different neuroblastoma cell lines (two with MNA and two without MNA) with palbociclib and performed RNA-seq for differential gene expression analyses. By comparing the genes most significantly downregulated by palbociclib among these cell lines via Venn program (Supplementary Fig. 13), we extracted 89 common genes (Supplementary Table 6). GSEA results showed that the 89-gene signature was significantly enriched within gene sets downregulated by KDM6B knockdown (Fig. 6c) or GSK-J4 treatment (Fig. 6d). Although single drug treatment effectively inhibited colony formation in a dose dependent manner, the combination of GSK-J4 and palbociclib did not enhance the inhibitory effect on colony formation of BE2C cells, as shown by colony formation and Bliss combination index (Fig. 6e, f), likely because both drugs target the same pathway. However, GSK-J4 greatly enhanced the effect of 17-DMAG, an HSP90 inhibitor (Supplementary Fig. 14a, b), and JQ-1, a bromodomain inhibitor that targets BRD4[63] (Supplementary Fig. 14c, d). Interestingly, we observed some degree of synergy of GSK-J4 and palbociclib, as well as 17-DMAG and JQ-1 in SK-N-AS cells (Supplementary Fig. 14e–g), the non-MYCN amplified cells with less impact on E2F pathways by KDM6B inhibition.

**Cancer cells with acquired CDK4/6 inhibitor resistance are less responsive to GSK-J4.** Upon mitogen stimulation, CDK4 and CDK6 phosphorylate the pRB protein, leading to release of E2F transcription factors from pRB for gene transcription[64]. CDK4/6 inhibitors including palbociclib showed promising efficacy in ER$^+$/HER2$^-$ breast cancer and have been approved for clinical use[65,66]. Over 100 clinical trials of CDK4/6 inhibitors for a variety of cancers are currently in progress[65]. However, acquired resistance to CDK4/6 inhibitors, as observed during breast cancer treatment, is expected and the mechanism involves amplification of *CDK6* or loss of function of *Rb1*[65,66]. To test if these mechanisms also diminish the effect of GSK-J4, we transduced the BE2C cells with lentiviral based CDK4 and CDK6 (Fig. 6g). Overexpression of CDK4 or CDK6 conferred expected resistance in neuroblastoma cells to palbociclib, as shown by colony formation and EC50 shift (Fig. 6h, i). Cells which overexpressed CDK4 or CDK6 also gained resistance to GSK-J4, although a higher concentration of GSK-J4 at 2.5 μM was still able to kill them (Fig. 6h, j). We then generated an *Rb1* knockout cell line for testing the response to GSK-J4 and palbociclib (Fig. 6k). As expected, loss of *Rb1* conferred resistance to palbociclib (Fig. 6l). Similarly, the *Rb1* knockout cells were also resistant to GSK-J4 treatment in comparison with the *Rb1* wildtype cells (Fig. 6m). Similar to the CDK4 or CDK6 overexpressing cells, the combination of palbociclib and GSK-J4 was not synergistic (Fig. 6m). These data further demonstrate that KDM6B acts on the same pathway as pRB-E2F in neuroblastoma.

**A gene signature targeted by KDM6B inhibition is associated with poor outcome.** To correlate the clinical relevance of target genes of KDM6B inhibition, we identified a 149-gene signature (Supplementary Table 7) that was commonly downregulated by GSK-J4 in BE2C and three other neuroblastoma cell lines (IMR5, LAN5, and SK-N-FI)[43]. Among these 149 genes, 85 were highly expressed in the high-risk neuroblastomas while 20 were highly expressed in the low-risk neuroblastomas (Supplementary Table 8). Based on the expression levels of these differentially expressed genes in neuroblastomas, they were categorized into 4 clusters (Fig. 7a). Cluster 3 and 4 neuroblastomas expressed

higher levels of the GSK-J4 signature, and were enriched with high-risk, high-stage and *MYCN*-amplified tumors. While patients in clusters 1 and 2 showed excellent event-free and overall survival, clusters 3 and 4 patients had a significantly worse outcome (Fig. 7b, c). No difference was observed between clusters 1 and 2. Cluster 4 showed a poorer survival than that of cluster 3, in line with the higher expression of GSK-J4 target genes in cluster 4. *MYCN* amplification (MNA) is a well-known poor prognostic risk factor in neuroblastoma. To investigate whether the correlation of the GSK-J4 signature with patient outcome was affected by MNA status, we compared the event-free survival of clusters 3 and 4 with MNA. The results showed that cluster 4, which had higher levels of GSK-J4 signature, was still correlated with a poorer event-free survival of patients with MNA although the overall survival was not statistically significant (Fig. 7d, f). However, in stage 4 patients, cluster 4 showed significantly worse outcome in both event-free and overall survival than that of cluster 3 (Supplementary Fig. 15). In high-risk patients, cluster 4 had a significantly worse outcome in event-free survival compared to cluster 3, but there was no difference in overall survival (Fig. 7e, g). These data indicate that, in stage 4 or MNA patients, higher expression levels of the GSK-J4 signature is a high-risk factor.

## Discussion

Despite one recent study suggesting that KDM6B might be tumor suppressive in neuroblastoma[67], we and others have found that KDM6B plays an important role in MYC-driven tumorigenesis and that pharmacologically targeting KDM6B by the small molecule GSK-J4 is therapeutically efficacious in multiple tumor models[14,27,68–71], including high-risk neuroblastoma[43]. However, the anticancer mechanisms of KDM6B inhibition are poorly understood. Here, we show that *KDM6B* is significantly over-expressed in human primary neuroblastoma in comparison with its paralogs *KDM6A* and *UTY*. Strikingly, the 22-kb genomic locus of *KDM6B* is heavily occupied by active gene transcription marks, including H3K4me1, H3K27Ac, BRD4 and RNA polymerase II binding while the *KDM6A* and *UTY* loci lack such histone modifications and transcription factor binding. These data explain why *KDM6B* is expressed at much higher levels than *KDM6A* and *UTY* in neuroblastoma. The long-distance broad marks at the KDM6B locus with H3K4me1 and H3K27Ac suggest *KDM6B* expression is under the control of a super-enhancer, a large cluster of transcriptional enhancers that drive expression of genes that define cell identity[72]. We further found that KDM6B is predominantly involved in regulation of MYC expression and downstream target genes of the pRB-E2F pathway. Genetic and pharmacologic inhibition of KDM6B consistently repressed the expression of *MYCN* and *C-MYC*, and the transcriptome of E2F target genes. These results support previous studies demonstrating that E2F and MYC are functionally associated[38,39], and that E2F drives MYC expression. MYC also regulates E2F expression and requires distinct E2F activities to induce S phase and apoptosis[73]. Moreover, a close association between E2F and MYC binding sites and their target genes has been observed[74]. Nevertheless, the impact of KDM6B inhibition on E2F pathways was more dramatic in MYCN-amplified neuroblastoma cells.

E2F deregulation in cancer often occurs through loss-of-function of the pRB tumor suppressor[75]. The E2F transcription factors (E2F1, E2F2, and E2F3a) bind at target gene promoters in complexes with their dimerization partner (DP1 or DP2) and pRB[36], which suppresses target gene transcription through the recruitment of chromatin modifiers and remodeling factors such as HDAC and EZH2[36,76]. During the G1 phase of the cell cycle, oncogenes such as Ras induce D-type cyclins to activate cyclin

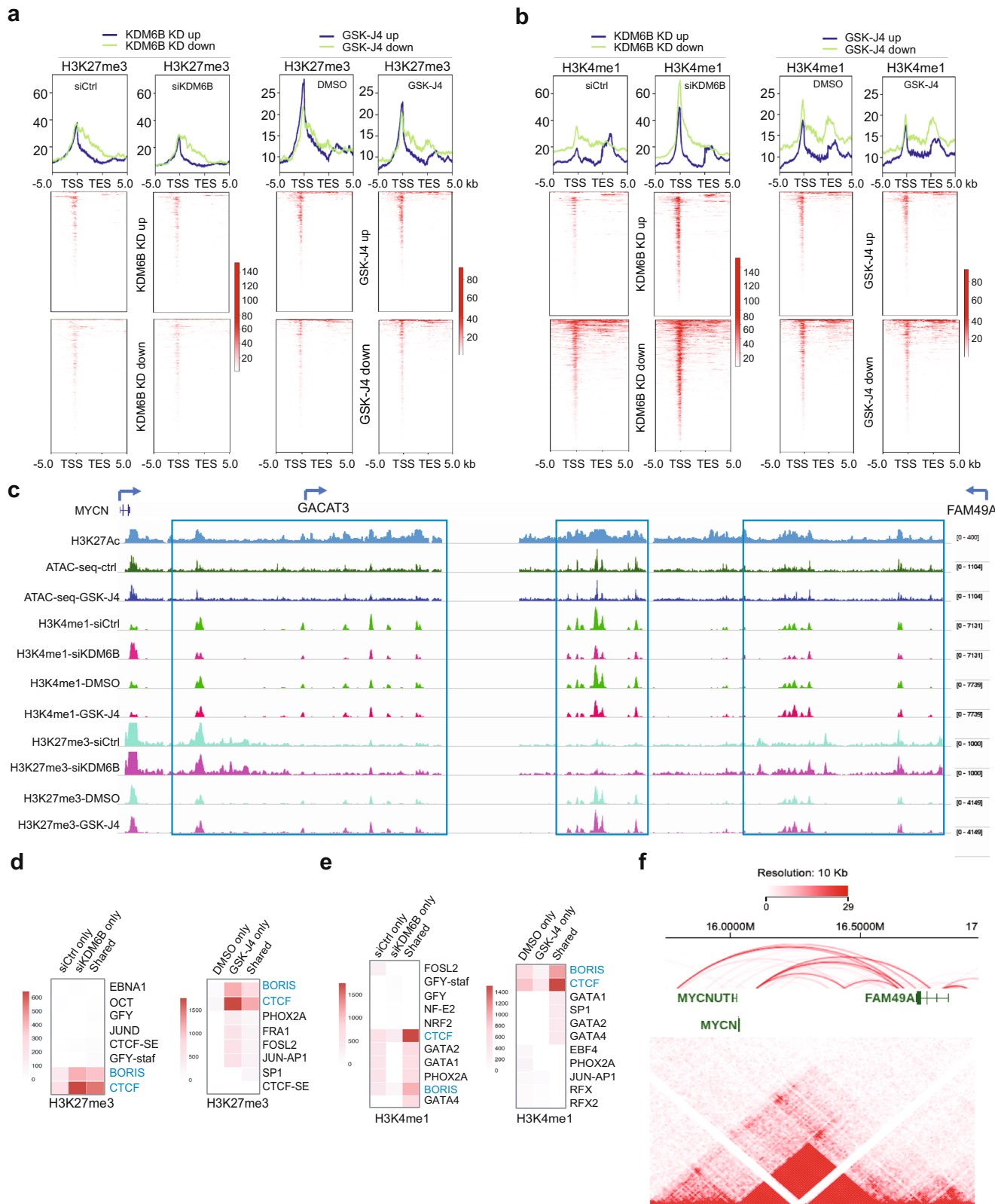

dependent serine/threonine kinases CDK4 and CDK6, which phosphorylate pRB, consequently leading to release of E2F transcription factors from pRB, allowing for cell proliferation[64]. While *Rb1* loss is very rare in neuroblastoma, several studies have shown that the cyclin D/CDK4/CDK6 pathway is hyperactive in neuroblastoma[77–82]. Genome-wide CRISPR and shRNA screening demonstrated that knockout of *CDK4 or CDK6* inhibited neuroblastoma cell proliferation/survival while *Rb1* knockout promoted cell proliferation (https://depmap.org/portal/). A majority of neuroblastoma models with MNA are sensitive to CDK4/6 inhibition[83]. These data indicate that the deregulated E2F pathway is essential to neuroblastoma. The genetic and pharmacologic KDM6B inhibition results in a predominant reduction of the E2F transcriptome and MYC, which may

**Fig. 5 KDM6B inhibition induces an increase of H3K27me3 but a decrease of the enhancer mark H3K4me1 at CTCF and BORIS binding sites. a, b** Heatmap indicating the CUT&Tag signal intensity of H3K27me3 (**a**) and H3K4me1 (**b**) around differentially expressed genes (from TSS-5kb to TES + 5 kb). CUT&Tag experiments were done with two biological replicates in KDM6B knockdown and GSK-J4 treatment, respectively. Differentially expressed genes were identified by cutoffs (log2FC > 0.5) from RNA-seq. TSS transcription start site, TES transcription end site. The *p*-value is obtained from the limma moderated *t*-statistic, Benjamini and Hochberg's method used to control the false discovery rate. **c** The signals of H3K27me3 and H3K4me1 (CUT&Tag), H3K27Ac (ChIP-seq)[109], and ATAC-seq between the genomic locus of *MYCN*, *GACAT3*, and *FAM49A*, snapshot using IGV program. **d, e** Motif analysis of H3K27me3 (**d**) and H3K4me1 (**e**) peaks by using HOMER known motif search. Top ten most significant motifs from each sub-group were visualized as a heatmap. The heatmap scale indicates −log$_{10}$(*p* value). *p*-Value obtained by HOMER scoring function of one-sided cumulative hypergeometric distribution. **f** The Hi−C data of BE2C neuroblastoma cells (data extracted from the St Jude Cloud) showing that *MYCN* and its adjacent *FAM49A* locus reside in the same topologically associated domain (TAD). The Arc indicates the chromatin interactions, which were generated from Jurkat ChIA-PET SMC1 (Mango) (https://proteinpaint.stjude.org/). Source data are provided as a "Source data" file.

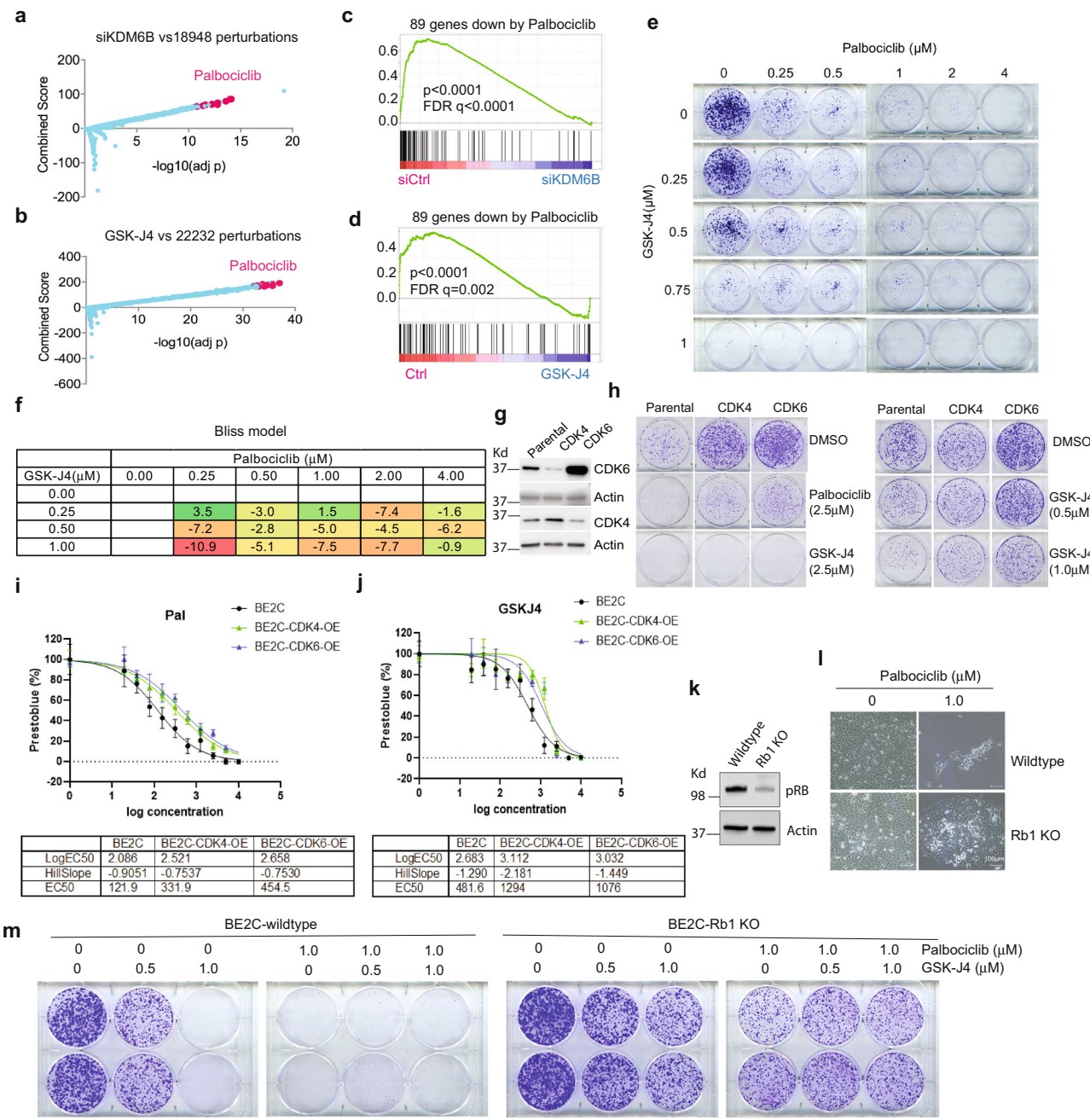

**Fig. 6 GSK-J4 induces a transcriptome that mimics CDK4/6 inhibition, and vice versa. a** Genes downregulated by KDM6B knockdown were compared with gene profiles induced with chemical compounds from Library of Integrated Network-Based Cellular Signatures (LINCS)[108]. The top chemical signature hits, which are basically overrepresented by palbociclib, are highlighted in pink dots. The y-axis indicates the combined score. The x-axis indicates adjusted p value. n = 18,947 for gene signatures. p Value is calculated from one-sided Fisher exact test. Combined score is computed by taking the log of p-value from the Fisher exact test and multiplying that by the z-score of the deviation from the expected rank. **b** Same analysis performed as in (**a**) for GSK-J4 treatment. n = 22231 for gene signatures. **c** The 89-gene signature derived from palbociclib treatment of 4 neuroblastoma cell lines was included in gene sets for GSEA analysis of KDM6B knockdown. p Value calculated by one-sided Fisher's exact test. The FDR is calculated by comparing the distribution of normalized enrichment scores from many different genesets. **d** The 89-gene signature derived from palbociclib treatment was included in gene sets for GSEA analysis of GSK-J4. p Value calculated by one-sided Fisher's exact test. The FDR is calculated by comparing the distribution of normalized enrichment scores from many different genesets. **e** BE2C cells were seeded with low numbers in six-well plate and treated with different concentrations of GSK-J4 or/ and palbociclib for 10 days. The cell colonies were stained with crystal violet. **f** Bliss index for the combination of GSK-J4 and palbociclib. Positive scores indicate synergy, negative scores indicate antagonism. **g** Western blot to assess the expression of CDK4 and CDK6 that were transduced into BE2C cells. The blots are representative of three independent experiments. **h** The BE2C parental and CDK4/6 overexpressing cells were seeded with low numbers in six-well plate and treated with different concentrations of GSK-J4 or palbociclib for 8 days. The cell colonies were stained with crystal violet. **i, j** BE2C and CDK4 or CDK6 overexpressing BE2C cells (BE2C-CDK4-OE, BE2C-CDK6-OE) were treated with different concentrations of palbociclib (**i**) or GSK-J4 (**j**). EC50 was calculated using Prestoblue assay. n = 4 for each dose. Data are represented as mean ± SD. **k** Western blot to assess the expression of pRB after inducible knockout in BE2C cells. The blots are representative of three independent experiments. **l** Photos taken under microscope (10×) show Rb1 knockout leads to resistance to palbociclib. Scale bar = 100 μM. The photos are representative of three independent experiments. **m** The BE2C wildtype and Rb1 knockout cells were seeded with low numbers in six-well plate and treated with different concentrations of GSK-J4 or/and palbociclib for 8 days. The cell colonies were stained with crystal violet. Source data are provided as a "Source data" file.

account for the therapeutic effect of KDM6B blockade in neuroblastoma. In addition, the transcriptome of KDM6B inhibition mimics the CDK4/6 inhibitor, palbociclib. When cells gained resistance to palbociclib by overexpressing CDK4/6 or by *Rb1* knockout, they also gained resistance to GSK-J4. Thus, these data further support that KDM6B regulates the CDK4/6-pRB-E2F pathway in neuroblastoma.

Intriguingly, while transcriptome, chemical genetics and ATAC-seq analyses show that KDM6B inhibition impacts MYCN and the E2F transcriptome, the loci of H3K27me3 alterations were not enriched with MYCN and E2F genes. As the majority of enhancer mark H3K4me1 and H3K27me3 peaks altered by KDM6B inhibition were located at distal non-promoter regions, one possible explanation is that KDM6B inhibition disrupted the long-range chromatin interaction between its targets and the regions with altered H3K27me3 and H3K4me1 because H3K4me1 functions to mediate the enhancer-promoter looping[56,57]. H3K4me1 is basically catalyzed by KMT2 family methyltransferases. Previous studies have shown that KMT2 members can form complexes with KDM6[84–86], suggesting a concerted mechanism for transcriptional activation in which cycles of H3K4 methylation by KMT2 are linked with the demethylation of H3K27[84–86]. The transcriptional factor motif analysis showed that CTCF and BORIS binding motifs are the major sites impacted by altered H3K27me3 and H3K4me1. KDM6B inhibition specifically leads to increased H3K27me3 at CTCF and BORIS binding sites while H3K4me1 at CTCF and BORIS binding sites are selectively reduced. These data suggest that increased H3K27me3 displaces H3K4me1 modifiers (highly likely the H3K4me1 methyltransferase KMT2) from CTCF and BORIS sites, consequently disrupting the enhancer activity. Previous studies have shown that enhancers, CTCF and H3K4me1 peaks overlap[87]. CTCF regulates the long-range chromatin interactions at enhancers. Thus, KDM6B inhibition may disrupt the chromatin interaction at its target genes. *MYCN* and many E2F target genes reside in the same topologically associated domains (TAD) as the regions with elevated H3K27me3 by KDM6B inhibition. TADs include chromatin loops that mediate promoter–enhancer contacts that regulate gene expression[58,59]. Our CUT&Tag data showed that the super-enhancer region of *MYCN*, which exhibits long-range chromatin interactions within a TAD domain, has the most significant changes of H3K27me3 and H3K4me1 by KDM6B knockdown and GSK-J4 treatment, in

line with the reduction of MYCN expression by KDM6B inhibition. In erythroid cells, long-range control of epigenetic regulation has been observed in that KDM6B is recruited to the enhancer regions to erase H3K27me3, consequently evicting the gene silencing PcG protein for a high rate of transcription[88]. During early differentiation steps, the embryonic stem cell factor Tbx3 associates with KDM6B at the enhancer element of the *Eomes* locus to allow enhancer-promoter interactions[89]. This spatial reorganization of the chromatin primes the cells to respond to Activin signaling. In neural stem cells, SMAD3 recruits KDM6B at the enhancers in response to TGF-β, for enhancer transcription[90]. Here we propose a model that, once stimulated by mitogens due to oncogenic activity, KDM6B is recruited to chromatin to maintain the low levels of H3K27me3 at the distal regulatory enhancer regions bound by CTCF and BORIS, which loops and physically interacts with E2F transcription factors that bind at the promoters of target genes, together with associated transcriptional machinery including RNA polymerase II and transcription mediators, driving the expression of *MYCN* and the E2F transcriptome (Fig. 8a). When KDM6B is inhibited, the H3K27me3 accumulates at the distal regions and disrupts the assembly of transcriptionally active interaction of promoter-enhancers, leading to the downregulation of *MYCN* and the E2F transcriptome (Fig. 8b).

Interestingly, expression of the PRC2 complex, which is a downstream target of pRB-E2F[60,61] and MYCN[28], was downregulated by KDM6B inhibition, in line with of the decrease of a number of H3K27me3 peaks and de-repression of a tumor suppressive program governed by EZH2[28]. The connection between KDM6B and EZH2 is interesting. Since EZH2 is a downstream target of E2F and MYCN, we assumed the reduction of EZH2 by KDM6B inhibition is through this mechanism. However, considering both proteins are epigenetic modifiers that confer cellular plasticity, we also speculate that cells strive for adaptation to the KDM6B inhibition by suppressing EZH2 in order to counterbalance the net increase of H3K27me3.

Neuroblastoma is a disease that arises as a result of blocked differentiation of NCCs during development[19,20]. EZH2 is required to maintain the undifferentiated state of neuroblastoma[28,91]. Thus, our data and that from other studies indicate that a network composed of MYCN, E2F, PRC2, and KDM6B regulates cell proliferation and differentiation of neuroblastoma (Fig. 8c), highlighting the importance of KDM6B in coupling the two essential

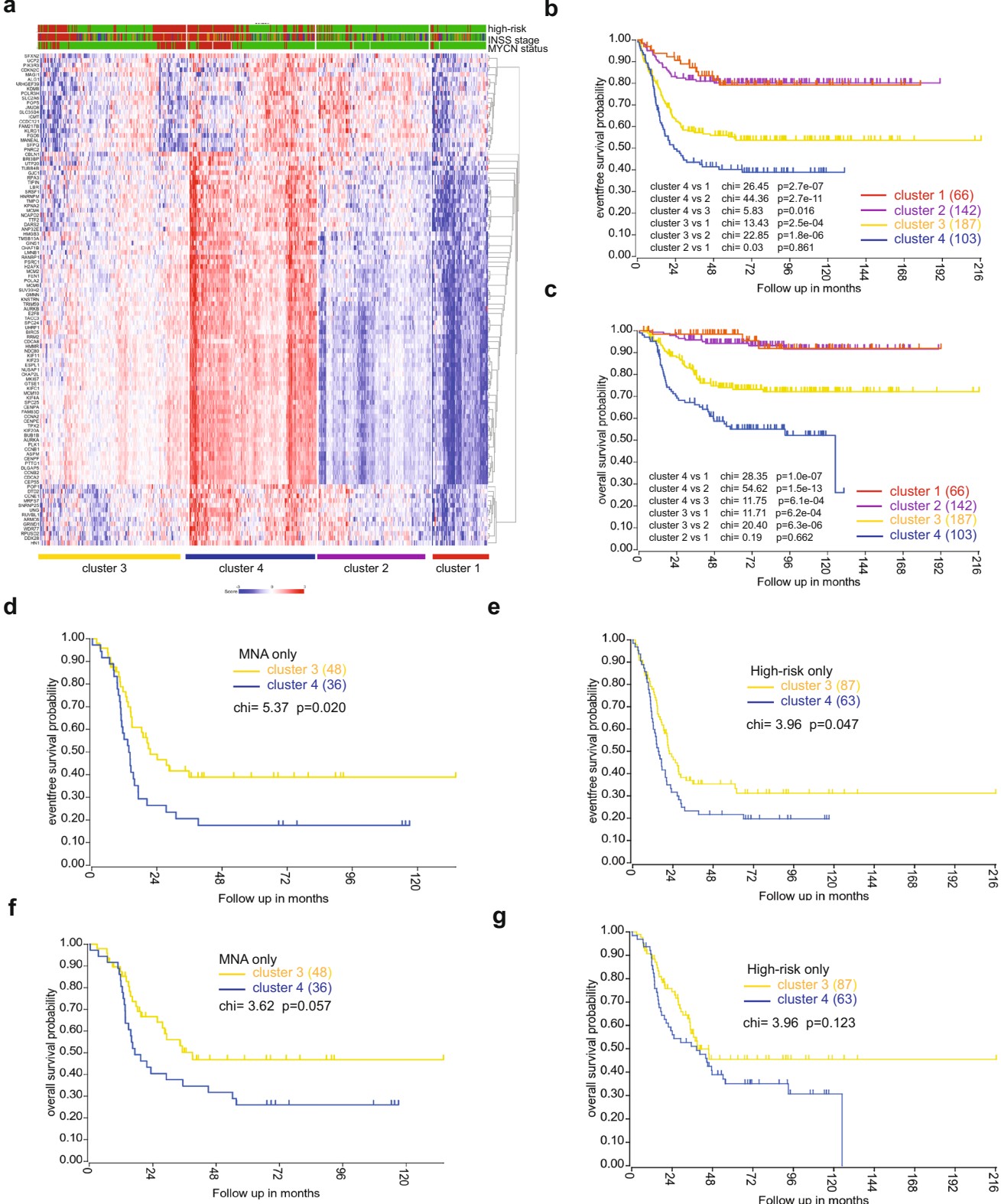

**Fig. 7 A gene signature targeted by KDM6B inhibition is associated with poor outcome. a** Totally, 149-gene signature was uploaded into R2 genomics analysis program (https://hgserver1.amc.nl/cgi-bin/r2/main.cgi) to find the differentially expressed genes in high-risk neuroblastoma in a cohort dataset (GSE49710) that has 498 neuroblastoma cases, followed by k-means cluster analysis. Risk (red = high risk; green = low risk); *MYCN* (red = amplification; green = non-amplification); stage (red = stage 4; blue = stage 4S; brown = stage 3; dark green = stage 2, green = stage 1). **b, c** Kaplan–Meier curve using the Log-Rank method shows the event-free survival and overall survival of four clusters that have differential expression levels of GSK-J4 signature. **d, f** Kaplan–Meier curve using the Log-Rank method shows the event-free survival and overall survival of clusters 3 and 4 with MNA. **e, g** Kaplan–Meier curve using the Log-Rank method shows the event-free survival and overall survival of clusters 3 and 4 with high-risk disease. Source data are provided as a "Source data" file.

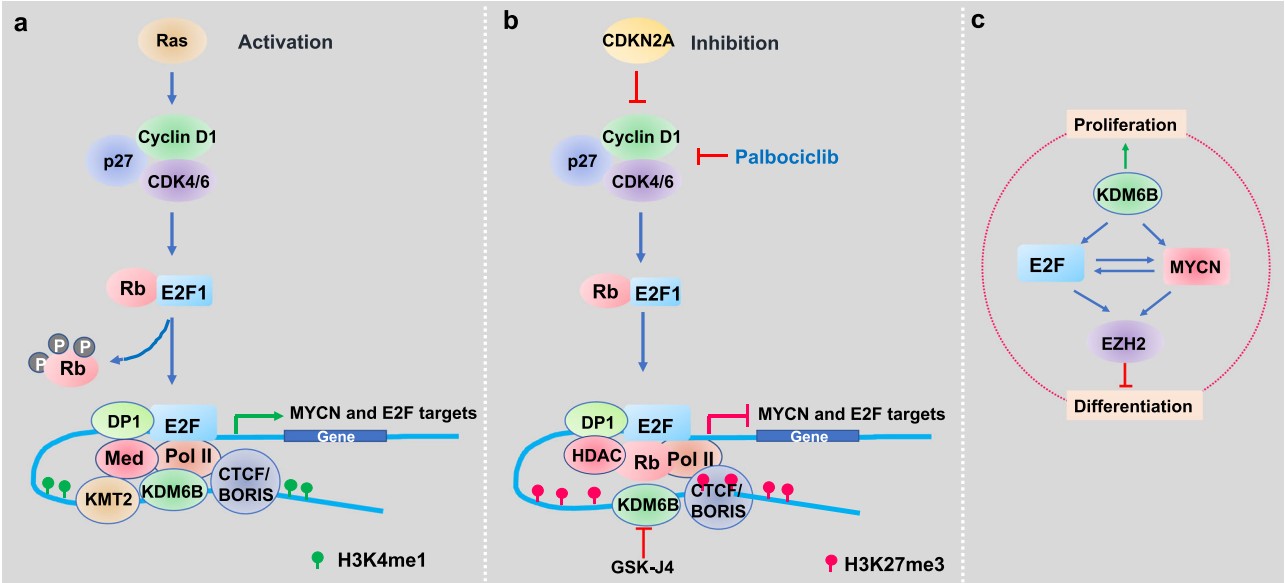

**Fig. 8 Working model of KDM6B inhibition in neuroblastoma. a** Upon stimulation by mitogens, KDM6B is recruited to chromatin to maintain the low levels of H3K27me3 at the distal regulatory enhancer regions marked by H3K4me1 overlapping with the CTCF-/BORIS-binding sites, which loops and physically interacts with E2F that binds at the promoter of target genes, together with associated transcriptional machinery including RNA polymerase II and mediators, driving the MYCN and E2F transcriptome. **b** When inhibited by CDK4/6 inhibitor, the inhibitory pRB complexes with HDAC to suppress gene transcription. When KDM6B is inhibited by GSK-J4, the H3K27me3 will accumulate at the distal regions to displace the H3K4me1 modifier KMT2 and evicts the transcription activators of promoter–enhancer, leading to the downregulation of MYCN and E2F transcriptome. **c** A network composed by MYCN, E2F, EZH2, and KDM6B regulates the cell proliferation and differentiation of neuroblastoma.

features of cancer cells. In summary, we have defined a chromatin-dependent mechanism of action of KDM6B inhibition that modulates the CDK4/6-pRB-E2F pathway in neuroblastoma. We also demonstrate that E2F target genes can act as biomarkers for sensitivity to KDM6B inhibitors in neuroblastoma and that this may well be predictive for other MYC-driven tumors.

## Methods

**Cell culture and reagents.** Neuroblastoma cell lines BE2(C) (ATCC), SIMA (DSMZ, Germany), SKNDZ (ATCC), SK-N-FI (ATCC), SK-N-AS (ATCC), SK-N-SH (ATCC), IMR32 (ATCC), CHLA20 (COG), CHLA15 (COG), KELLY (ECACC), SKNBE2 (COG), NB-1691 (Peter Houghton), HS68 (ATCC) were cultured in standard RPMI media supplemented with 10% fetal bovine serum (FBS) (Sigma), 1% L-glutamine (MediaTech), at 37 °C in 5% CO₂. Colorectal cancer cell lines HCT116, and isogenic p53⁻/⁻ HCT116 cells were kindly provided by Bert Vogelstein (Baltimore, MD), osteosarcoma cell line U2OS, lung cancer cell lines A549 purchased from ATCC, and they were cultured in standard DMEM media supplemented with 10% FCBS, 1% L-glutamine (MediaTech), at 37 °C in 5% CO₂. Cell lines were validated by short tandem repeat using Promega PowerPlex 16 HS System once per month. The polymerase chain reaction (PCR)-based method was used for detection of Mycoplasma with LookOut Mycoplasma PCR Detection Kit (Sigma) and JumpStart Taq DNA Polymerase (Sigma) once per month to ensure cells were mycoplasma negative. GSK-J4, Palbociclib, 17-DMAG, JQ-1 were purchased from Selleckchem. KDM5-C70 was purchased from Xcess Biosciences Inc.

**AlphaLISA.** Materials: GSK J1 is purchased from Tocris (Bristol, United Kingdom, Catalog number 4593). 2,4-Pyridine Dicarboxylic Acid (2,4-PCA) is purchased from Acros Organics (New Jersey, Catalog number 101860010). JIB-04 is purchased from Sigma-Aldrich (St. Louis, MO, Catalog number SML0808). Daminozide is purchased from Tocris (Bristol, United Kingdom, MO, Catalog number 4684). AlphaLISA anti-mIgG acceptor beads from PerkinElmer (Santa Clara, CA, Catalog number AL105C). AlphaLISA anti-rIgG acceptor beads from PerkinElmer (Santa Clara, CA, Catalog number AL104C). AlphaScreen Streptavidin-conjuagated donor beads from PerkinElmer (Santa Clara, CA, Catalog number 6760002). Primary antibody 5 from BPS (Catalog number 52140E). Primary antibody 6 from BPS (Catalog number 52140 F). Primary antibody 13-4 from BPS (Catalog number 52140M4). Primary antibody 16-2 from BPS (Catalog number 52140P-2). Primary antibody 17-4 from BPS (Catalog number 52140Q4). Biotinylated histone H3 peptide substrate (KDM4A) BPS (Catalog number 79841). Biotinylated histone H3 peptide substrate (KDM5A) BPS (Catalog number 79840). Biotinylated histone H3 peptide substrate (KDM2A) BPS (Catalog number 79843). Biotinylated histone H3 peptide substrate (KDM6A and KDM6B) BPS (Catalog

number 79841). Biotinylated histone H3 peptide substrate (KDM3A) BPS (Catalog number 79841). 4× HDM Assay Buffer 2 from BPS (Catalog number 52407). 4× HDM Assay Buffer 3 from BPS (Catalog number 52408). 4× HDM Assay Buffer 4 from BPS (Catalog number 52409). 4× HDM Assay Buffer 5 from BPS (Catalog number 79847). 4× Detection buffer from BPS (Catalog number 52301).

Assay conditions: All of the enzymatic reactions were conducted in duplicate at room temperature for 60 min in a 10 µl mixture containing assay buffer, histone H3 peptide substrate, demethylase enzyme, and the test compound. These 10 µl reactions were carried out in wells of 384-well Optiplate (PerkinElmer). The dilution of the control compounds was first performed in 100 % DMSO. Each intermediate compound dilution (in 100% DMSO) will then get directly diluted 30× fold into assay buffer for 3.3× conc. (DMSO). Enzyme only and blank only wells have a final DMSO concentration of 1%. From this intermediate step, 3 µl of compound is added to 4 µl of demethylase enzyme dilution is incubated for 30 min at room temperature. After this incubation, 3 µl of peptide substrate is added. The final DMSO concentration is 1%. After enzymatic reactions, 5 µl of anti-Mouse Acceptor beads (PerkinElmer, diluted 1:500 with 1× detection buffer) or 5 µl of anti-Rabbit Acceptor beads (PerkinElmer, diluted 1:500 with 1× detection buffer) and 5 µl of Primary antibody (BPS, diluted 1:200 with 1× detection buffer) were added to the reaction mix. After brief shaking, plate was incubated for 30 min. Finally, 10 µl of AlphaScreen Streptavidin-conjugated donor beads (Perkin, diluted 1:125 with 1× detection buffer) were added. In 30 min, the samples were measured in AlphaScreen microplate reader (EnSpire Alpha 2390 Multilabel Reader, PerkinElmer).

Data analysis: Enzyme activity assays were performed in duplicates at each concentration. The A-screen intensity data were analyzed and compared. In the absence of the compound, the intensity (Ce) in each data set was defined as 100 % activity. In the absence of enzyme, the intensity (C0) in each data set was defined as 0% activity. The percent activity in the presence of each compound was calculated according to the following equation: % activity = (C − C0)/(Ce − C0), where C = the A-screen intensity in the presence of the compound. The values of % activity were plotted on a bar graph.

**SDS-PAGE and Western blotting.** For western blotting, samples were mixed with calculated volume of 2× sample buffer (1 M TRIS/HCl, 10% SDS, 0.1% bromophenol-blue, 10% b-mercaptoethanol, 10% glycerol), and heated for 25 min at 96 °C. Proteins were resolved on SDS-PAGE and transferred onto PVDF membrane (Immobilon-P, Millipore). After being incubated with the primary antibody, horseradish peroxidase-(HRP) conjugated secondary antibody (Novex, Life technologies) at 1:5000 was used for 1 h incubation. The signals were detected by chemiluminescence (ECL, Thermo scientific). Antibodies including H3K4me1(Abcam Cat# ab8895, RRID:AB_306847), H3K4me3 (RevMab Biosciences Cat# 31-1226-00, RRID:AB_2783580), H3K27ac (Active Motif Cat# 39133, RRID:AB_2561016), H3K27me3 (Cell Signaling

Technology Cat# 9733, RRID:AB_2616029), Total H3 (Cell Signaling Technology, Cat#4499, RRID: AB_10544537), CDK4 (Cell Signaling Technology Cat# 12790, RRID:AB_2631166), CDK6 (Cell Signaling Technology Cat# 13331, RRI-D:AB_2721897), β-ACTIN (Sigma-Aldrich Cat# A1978, RRID:AB_476692), GAPDH (Cell Signaling Technology Cat# 5174, RRID:AB_10622025), pRB (4H1) (Cell Signaling Technology Cat# 9309, RRID:AB_823629), MYCN (Cell Signaling Technology Cat# 9405, RRID:AB_10692664) or (Santa Cruz Biotechnology Cat# sc-53993, RRI-D:AB_831602), C-MYC (Cell Signaling Technology Cat# 13987, RRID:AB_2631168), KDM6B (Abclonal, A12763), H3K9me3 (Active Motif, 39161), H3K36me3 (Abcam, ab9050), Secondary HRP-conjugated goat anti-mouse (Thermo Fischer Scientific Cat#31430 RRID:AB_228307), and Secondary HRP-conjugated goat anti-rabbit (Thermo Fischer Scientific Cat#31460 RRID:AB_228341) were used for western blot.

**Crystal violet staining**. After removing media, cells were washed with Dulbecco's phosphate buffered saline without calcium or magnesium (DPBS, Lonza) and treated with 4% Formaldehyde in PBS (PFA) for 20 min. Once PFA was removed, cells were stained with 0.1% crystal violet stain for 1 h.

**Synergy assay**. BE2C and SK-N-AS cells were seeded in 96-well plates (3000 cells per well). After 24 h, cells were treated with GSK-J4 (0, 0.25, 0.5, 1 μM, for BE2C, 0, 0.5, 1, 2.5 μM for SK-N-AS) and palbociclib (0, 0.25, 0.5, 1.0, 2.0, 4.0 μM) or JQ1 (0, 0.0625, 0.125, 0.25, 0.5, 1.0 μM) or 17-DMAG (0, 0.0625, 0.125, 0.25, 0.5, 1.0 μM) in a 4 × 6 matrix. Every combination treatment was performed in quadruplicate. Cells were treated for 5 days, and cell viability was determined using the Prestoblue assay (Invitrogen, A-13262). Cell viability for each treatment was normalized against the control group. A Bliss independence model was used to evaluate combination effects. Percentage over the Bliss score index was calculated with the equation $(A + B) - A \times B$, in which A and B are the percentage of growth inhibitions induced by agents A and B at a given dose, respectively. The difference between the Bliss expectation and the observed growth inhibition induced by the combination of agent A and B at the same dose is the Bliss excess.

**Cell viability assay**. BE2C, and CDK4 or CDK6 overexpressing BE2C (BE2C-CDK4-OE, BE2C-CDK6-OE) cells were seeded in 96-well plates at 3000 cells per well. After 24 h, cells were treated with GSK-J4 (0, 0.02, 0.04, 0.08, 0.16, 0.3125, 0.625, 1.25, 2.5, 5.0, and 10 μM) and palbociclib (0, 0.02, 0.04, 0.08, 0.16, 0.3125, 0.625, 1.25, 2.5, 5.0, and 10 μM). Cells were treated for 5 days. Cell viability was determined using the Prestoblue assay (Invitrogen, A-13262). Cell viability for each treatment was normalized against no treatment well. EC50 was determined with GraphPad Prism equation $Y = 100/(1 + 10^{((LogEC50-X)*HillSlope)})$, EC50 is the concentration that gives a 50% response. HillSlope represents the steepness of the curve.

**Small interfering RNA transfection**. Small interfering RNAs (siRNA) were transfected into subconfluent cells using Lipofectamine RNAiMax (Invitrogen) according to manufacturer's instructions. Non-Targeting siRNA#1 (Dharmacon, D-001810-0105) used as siRNA control. The siRNA oligos for KDM6B have sequences as follows: KDM6B#1, 5-GGAAUGAGGUGAAGAACGU-3, KDM6B#2, 5-GGAGACCUCGUGUGGAUUA-3, KDM6B#31, 5-GCAUCUAUCUGGAGAGCAA-3, KDM6B#33, 5-GGAAGAGGAACAGCAACA-3. KDM6B-3'UTR, 5-AGAAAGAACUAUGAGGAAAUU-3.

**CRISPR knockout of Rb1**. The Rb1 CRISPR plasmid with gRNA sequence 5-GCTCTGGGTCCTCCTCAGGA-3 (TLCV2-RB1, Addgene#87836) was purchased from Addgene. Plasmids were maxipreped by using NucleoBond Xtra EF kits (Takara Bio USA, 740424-50) according to manufacturer's protocol. Lentivirus was produced by transient transfection of PEI-pro DNA complex (6 μg of TLCV2-RB1, 3 μg of 1-1r, 1 μg RTR, 1 μg of VSVg with 22 μl of PEI pro in 400 μl of DMEM medium) with $5 \times 10^6$ HEK293T cells in 10 ml complete medium (DMEM, 100 U/mL penicillin/streptomycin, 1× L-glutamine and 10% FBS) in a 10 cm dish. Virus supernatant was collected every 8–12 h for 3 days, which were passed through a 0.45 μm filter and concentrated by ultracentrifuge at 50,000g for 1.5 h at 4 °C. The TLCV2-RB1 virus particles were added to BE2C cells with polybrene to final concentration of 8 μg/ml. Puromycin (2.5 μg/ml in complete medium) selection were performed in the next day after virus transduction. To generate Rb1 knockout in BE2C, BE2C-Rb1 cells were treated with 1 μM of doxycycline for 72 h, then sorted for green fluorescent protein (GFP) positive cells to enrich for RB1 knockout. The sorted cells were expanded without doxycycline in RPMI 1640 media (Corning, 10-040-CM) supplemented with 100 U/mL penicillin/streptomycin (Gibco, 15140122), and 10% FBS (Sigma-Aldrich, F2442).

**Generation of KDM6B, CDK4, and CDK6 overexpressing cell lines**. The CDK4 plasmid (pHAGE-CDK4, Addgene#116724) and CDK6 plasmid (pHAGE-CDK6, Addgene#116725) were purchased from Addgene. The plasmid maxiprep and lentiviral packaging were followed by the same protocol as described in Rb1 knockout. To overexpress CDK4 and CDK6 in BE2C cells, the pHAGE-CDK4 or pHAGE-CDK6 viral particles were transduced to BE2C cells and sorted for GFP positive ones. To overexpression KDM6B, MSCV-JMJD3 (Addgene#21212) were

packaged into retroviral particles, which were transduced into BE2C cells and selected with puromycin for stable expression.

**RNA-seq and microarray**. Total RNA was extracted from cells by using RNeasy Mini Kit (cat. # 74104) from QIAGEN. Paired-end sequencing was performed using the High-Seq platform with 100 bp read length. Reads were aligned to the human GRCh37-lite using SJCRH's Strongarm pipeline. Counts per gene were obtained using htseq-count version 0.6.1 with Gencode vM5 level 1 and 2 gene annotations. Counts were normalized with VOOM and analyzed with LIMMA within the R statistical environment. Significance was defined as having a false discovery rate (FDR) < 0.05. VOOM normalized counts were analyzed with GSEA[92]. For Affymetrix microarray, after quality control with Agilent RNA analyzer, RNA was subjected to hybridization using an Affymetrix Clariom S human array. Differential gene expression was analyzed by *t* test using the Differential Expression Analysis module at GenePattern server (http://genepattern.broadinstitute.org/gp/pages/protocols/DiffExp.html).

**Assay for transposase-accessible chromatin using sequencing (ATAC-seq)**. Library preparations for ATAC-seq were based on the protocol with minor modifications[47,93]. Briefly, fresh cultured BE2C cells (100,000 per sample) with or without 2.5 μM of GSK-J4 treatment were harvested and washed with 150 μl cold Dulbecco's Phosphate-Buffered Saline (DPBS) containing protease inhibitor (PI). Nuclei were collected by centrifuging at 500g for 10 min at 4 °C after cell pellets were resuspended in lysis buffer (10 mM Tris-Cl pH 7.4, 10 mM NaCl, and 3 mM MgCl2 containing 0.1% NP-40 and PI). Nuclei were incubated with Tn5 transposon enzyme in transposase reaction mix buffer (Illumina) for 30 min at 37 °C. DNAs were purified from transposition sample by using Min-Elute PCR purification kit (Qiagen, Valencia, CA) and measured by Qubit. Polymerase chain reaction (PCR) was performed to amplify with High-Fidelity 2× PCR Master Mix [72 °C/5 min + 98 °C/30 s + 12 × (98 °C/10 s + 63 °C/30 s + 72 °C/60 s) + 72 °C/5 min]. The libraries were purified using Min-Elute PCR purification kit (Qiagen, Valencia, CA). ATAC-seq libraries followed by pair-end sequencing on HiSeq4000 (Illumina) in the Hartwell Center at St. Jude Children's Research Hospital.

**ATAC-seq data analysis**. A 2 × 100 bp paired-end reads obtained from all samples were trimming for Nextera adapter by cutadapt(version 1.9, paired-end mode, default parameter with " -m 25 -O 6")[94] and aligned to human genome hg19(GRCh37-lite) by BWA (version 0.7.12-r1039, default parameter)[95], duplicated reads were then marked with Picard(version 2.6.0-SNAPSHOT)[96] and only non-duplicated proper paired reads have been kept by samtools (parameter "-q 1 -F 1804" version 1.2)[97]. After adjustment of Tn5 shift (reads were offset were offset by +4 bp for the sense strand and −5 bp for the antisense strand) we separated reads into nucleosome free, mononucleosome, dinucleosome, trinucleosome[93] by fragment size and generated bigwig files by using the center 80 bp of fragments and scale to 20 M nucleosome free reads. We observed reasonable nucleosome free peaks and pattern of mono-, di-, tri-nucleosome on IGV (version 2.4.13)[98]. Next we merged each 2 replicates to enhance peak calling on nucleosome free reads by MACS2 (version 2.1.1.20160309 default parameters with "-extsize 200 -nomodel")[99], all cell types have more than 20 M nucleosome free reads after merge so we consider all important nucleosome free regions were called. To assure the replicability, we first merge peaks from different treatments to create a set of reference chromatin accessible regions. We then counted nucleosome free reads from each of samples overlap the reference regions by bedtools (v2.24.0)[100]. Data reproducibility between replicates is assessed by correlation analysis of mapped read counts across the genome. We conclude the reproducibility is good since spearman correlation coefficient between replicates are larger than between samples from different groups. To find the differential accessible regions, we first normalized raw nucleosome free reads counts used trimmed mean of *M*-values normalization method and applied Empirical Bayes Statistics test after linear fitting from voom packge (R 3.23, edgeR 3.12.1, limma 3.26.9)[101]. FDR-correct *p*-value 0.05 and fold change > 2 were used as cutoff for DARs while *p*-value > 0.5 and fold change < 1.05 were used for control regions. To find the Transcription factor enriched for DARs, we scanned the TRANSFAC[102] motif database using FIMO (parameter "-motif-pseudo 0.0001 -thresh 1e-4") from MEME suite (v4.11.3)[103], then for each motif, we counted how many DARs or control regions have the motif matches and using Fisher exact test to estimate their enrichment over the background (DAR or control regions do not have the motif matches).

**CUT&Tag and analysis**. CUT&Tag for KDM6B knockdown and GSK-J4 treatment was prepared by following the protocol (Kaya-Okur et al. 2019 and https://www.protocols.io/view/bench-top-cut-amp-tag-bcuhiwt6?step=1) with minor modifications. Approximately, 500,000 BE2C cells were treated with siRNAs (siCtrl or siKDM6B) or compounds (DMSO 0.1% or GSK-J4 2.5 μM) for 72 h, followed by washing with wash buffer (20 mM HEPES pH 7.5; 150 mM NaCl; 0.5 mM Spermidine; 1× Protease inhibitor cocktail). Nuclei were isolated with cold NE1 buffer (20 mM HEPES–KOH, pH 7.9; 10 mM KCl 0.1%; Triton X-100; 20% Glycerol, 0.5 mM Spermidine; 1× Protease Inhibitor) for 10 min on ice. Nuclei were collected by 600 × g centrifuge and resuspended in 1 ml washing buffer containing with 10 μL of activated concanavalin A-coated beads (Bangs laboratories, BP531) at RT

for 10 min. Bead-bound nuclei were collected with placing tube on magnet stand and removing clear liquid. The nuclei bound with bead were resuspended in 50 μL Dig-150 buffer (20 mM HEPES pH 7.5; 150 mM NaCl; 0.5 mM Spermidine; 1× Protease inhibitor cocktail; 0.05% Digitonin; 2 mM EDTA) and incubated with a 1:50 dilution of H3K4me1 (Abcam, ab8895) and H3K27me3 (CST, 9733S) antibodies overnight at 4 °C. The unbound primary antibody was removed by placing the tube on the magnet stand and withdrawing the liquid. The primary antibody bound nuclei bead was mixed with 100 μL of Dig-150 buffer containing guinea pig anti-Rabbit IgG antibody (Antibodies, ABIN101961) in 1:100 dilution for 1 h at RT. Beads bound nuclei were washed using the magnet stand 3× for 5 min in 1 mL Dig-150 buffer to remove unbound antibodies. A 1:100 dilution of pA-Tn5 adapter complex was prepared in Dig-300 buffer (20 mM HEPES, pH 7.5, 300 mM NaCl, 0.5 mM Spermidine, 0.05% Digitonin, 1× Protease inhibitor cocktail). After removing the liquid on the magnet stand, 100 μL mixture of pA-Tn5 and Dig- 300 buffer was added to the nuclei bound beads with gentle vortex and incubated at RT for 1 h. After 3× 5 min in 1 mL Dig-300 buffer to remove unbound pA-Tn5 protein, nuclei were resuspended in 250 μL Tagmentation buffer (10 mM MgCl$_2$ in Dig-300 buffer) and incubated at 37 °C for 1 h. Ten microlitre of 0.5 M EDTA, 3 μL of 10% SDS and 2.5 μL of 20 mg/mL Proteinase K were added to stop tagmentation and incubated at 55 °C for 1 h. DNA was then precipitated by phenol/chloroform/ isoamylalcohol followed by ethanol precipitation with glycogen and then dissolved in water. Sequencing libraries were prepared using NEBNext HiFi 2× PCR Master Mix (NEB, M0541L) according to the manufacturer's instructions. The PCR products were cleaned up with SPRIselect beads and quantified using Qubit dsDNA HS assay kit (Agilent Technologies). The libraries were sequenced on a HiSeq2500 with paired-end 50-bp reads (Illumina).

*Mapping reads and peak calling.* The Cut&Tag raw reads were aligned to the human reference genome (hg19) using BWA (version 0.7.12; BWA aln+sampe). Duplicate reads were marked and removed by Picard (version 1.65). Only properly paired uniquely mapped reads with a fragment size of 150–2000bp were extracted by samtools (version 1.3.1 parameters used were -q 1 -f 2 -F 1804) for calling peaks and generating bigwig file. Narrow peaks were called by MACS2 (version 2.2.7.1) with parameters of " -t cut_tag_file -q 0.05 -f BED -keep-dup all" for CUT&Tag data).We used SICER (version 1.1, with parameters of redundancy threshold 1, window size 200 bp, effective genome fraction 0.86, gap size 600 bp, FDR 0.00001 with fragment size defined above) for calling broad enriched regions. Unmapped reads were further aligned to E.Coli genome as a spike-in control to calculated ChIP-Rx scaling factor[104,105]. The Cut&Tag library size of each sample was adjusted by E.Coli spike-in scaling factor and then used to normalize the read count for the downstream differential analysis, IGV bigwig and heatmap visualization.

*Visualization.* We used genomeCoverageBed (bedtools 2.25.0) to produce genome-wide coverage in BEDGRAPH file and then converted it to a bigwig file by bedGraphToBigwig. The bigwig signals were adjusted by E.Coli spike-in scaling factor and scaled to 15 million reads to allow comparison across samples. To show average of several replicates as a single track in the browser, the bigwig files were merged to a single average bigwig file using UCSC tools bigWigtoBedGraph, bigWigMerge and bedGraphToBigWig. The Integrated Genomics Viewer (IGV 2.4.13) was used for visual exploration of data. DeepTools (version 2.3)[106] was used to plot heatmap.

*Differential analysis.* Cut&Tag raw read counts were reported for each region/ each sample using bedtools 2.25.0. Raw read counts were Voom normalized and statistically contrasted using the R (version 3.5.1) packages limma and edgeR (version 3.16.5) for CPM calculation and differential analysis. An empirical Bayes fit was applied to contrast treated samples to control samples and to generate log fold changes, p values and FDRs for each peak region.

*Peak overlap and annotation.* Peak regions were defined to be the union of peak intervals in replicates from control or treated cells respectively. For peak overlap analysis, mergeBed (bedtools version 2.25.0) was used to combine overlapping regions from multiple peak sets into a new region and then a custom script was used to summarize common or distinct peaks and visualize in a venn diagram. Promoter regions was defined as the regions 1.0 kb upstream and 1.0 kb downstream of the TSSs based on the human RefSeq annotation (hg19). Genomic feature annotation of peaks was done by annotatePeaks.pl, a program from the HOMER suite (v4.8.3, http://homer.ucsd.edu/homer/).

*Motif analysis.* Adjacent histone mark peaks (within a distance of 100 bp) from both control and treated samples were merged first and then separated to 3 subgroups including control-only, treatment-only and shared peaks (at least 1 bp overlap) using bedtools (version 2.25.0). The regions of peak center ± 100 bp were used for HOMER known motif search. Top ten most significant motifs from each sub-group were visualized as a heatmap.

**Hi–C data mining.** To view the 3D chromatin conformation of *MYCN* and *E2F8* gene in BE2C, we mined the Hi-C data, of BE2C, KELLY, and SK-N-AS cells, which were downloaded from St. Jude Cloud (https://www.stjude.cloud/)[107]. The Arc TRAC data showing the chromatin interactions, which were generated from Jurkat ChIA-PET SMC1 (Mango), were downloaded from St Jude Protein Paint program under St. Jude Cloud (https://proteinpaint.stjude.org/).

**LINCS analysis.** We had two sets of gene expression data: microarray and RNA-seq for siKDM6B and GSK-J4 treatment. While the pathway analysis showed very similar results for each, we used them in different purposes. Because Lincs used a limited number of genes in charactering chemical compounds signatures, we chose our microarray data to compare since RNA-seq data gave rise to a large amount of gene expression data that were not covered by Lincs chemical signatures. We first identify differentially expressed genes in microarray data using GenePattern program (https://cloud.genepattern.org/gp/pages/index.jsf). Then we chose the downregulated genes (with logFC ≥ 0.7) by KDM6B knockdown or GSK-J4 treatment to compare with the chemical signatures in LINCS database (http://www.lincsproject.org/LINCS/tools)[108]. The datasets from LINCS L1000 Chem Pert down were downloaded and analyzed using PRISM program.

**The Cancer Therapeutics Response Portal (CTRP) analysis.** Cancer Therapeutic Response Portal (https://portals.broadinstitute.org/ctrp.v2.1/) correlated the sensitivity patterns of 481 compounds including GSK-J4 with 19,000 basal transcript levels across 823 different human cancer cell lines and identified selective outlier transcripts[44,45], which allowed to interrogate whether KDM6B inhibitor was correlated with specific transcripts. We chose the features that show the correlation of drug sensitivity with gene expression or copy number of quired genes such as E2F8.

**Kaplan–Meier analysis.** A 149-gene signature commonly downregulated by GSK-J4 in BE2C and three other neuroblastoma cell lines (IMR5, LAN5 and SK-N-FI)[43] was uploaded into R2 genomics analysis and visualization program (https://hgserver1.amc.nl/cgi-bin/r2/main.cgi) to find the differentially expressed genes in high-risk neuroblastoma in dataset (GSE49710) that has 498 cases, followed by k-means cluster analysis. The 4 clusters were stored as track for Kaplan–Meier curve analysis using the Log–Rank method for event-free survival and overall survival.

**Statistical analysis.** To determine statistical significance, the unpaired, two-tailed Student t test was calculated using the t test calculator available on GraphPad Prism 9.0 software. A p value of less than 0.05 was considered statistically significant.

**Reporting summary.** Further information on research design is available in the Nature Research Reporting Summary linked to this article.

## Data availability

Source data are provided with this paper. The RNA-seq, ATAC-seq and Cut&Tag raw data generated in this study have been deposited to the Gene Expression Omnibus-GEO (NCBI) as Super Series under accession number GSE149539 (GSE149537 [RNA-seq], GSE149519 [ATAC-seq], GSE182884 [CUT&TAG]). The microarray data generated in this study have been deposited to the Gene Expression Omnibus-GEO (NCBI) under accession number GSE150045. The uncropped western blot images are present in a Source Data file. Databases/datasets were used in the study including Library of Integrated Network-based Cellular Signatures (LINCS) (https://maayanlab.cloud/ Enrichr/), The Cancer Therapeutics Response Portal (CTRP) (https:// portals.broadinstitute.org/ctrp.v2.1/), R2: Genomics Analysis and Visualization Platform (https://hgserver1.amc.nl/cgi-bin/r2/main.cgi), Cancer Cell Line Encyclopedia-CCLE (https://sites.broadinstitute.org/ccle/), St Jude Pecan Portal (https://pecan.stjude.cloud/; https://pecan.stjude.cloud/proteinpaint/study/mycn_nbl_2018), CellMinerCDB (https:// discover.nci.nih.gov/rsconnect/cellminercdb/). TRANSFEC (https://genexplain.com/ transfac/). The authors declare that other data supporting the findings of this study are provided in the Supplementary information/Data/Source Data files. Source data are provided with this paper.

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

## Acknowledgements

This work was partly supported by American Cancer Society-Research Scholar (130421-RSG-17-071-01-TBG, J.Y.) and National Cancer Institute (1R01CA229739-01, J.Y.). Xinyu von Buttlar and Victoria Jones supported in part by R25CA23944 from the National Cancer Institute. The content is solely the responsibility of the authors and does not necessarily represent the official views of the National Institutes of Health.

## Author contributions

A.D'.O., J.F., S.S., A.M., V.J., X. von B., B.C., D.H. and A.A.-Z. performed the experiments. B.X. and H.J. analyzed the sequencing data. J.Y. and A.M.D. conceived the project. J.Y. wrote the paper with input from J.S., A.J.M. and A.M.D.

## Competing interests

The authors declare no competing interests.

## Additional information

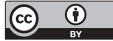

