## [Peer Review File · Nature Communications]

KDM6B promotes activation of the oncogenic CDK4/6-pRB-E2F pathway by maintaining enhancer activity in MYCN-amplified neuroblastomaEditorial Note: Parts of this Peer Review File have been redacted as indicated to remove third-party material where no permission to publish could be obtained.

Reviewers' comments:

Reviewer #2 (Remarks to the Author):

To investigate the role of KDM6B in regulating neuroblastoma. The authors present a wide swath of data to show that KDM6B is upregulated in neuroblastoma, has open and transcribed chromatin and that this gene plays a role in E2F and MYCN. The use both genetic and pharmacologic mechanisms of inhibition to demonstrate the robust impact of this gene and its inhibition. These data certainly do contribute to our understanding of the epigenetic regulation of neuroblastoma and the role of histone modifications. However, there are some limitations in this manuscript, particular the heavy reliance on what appears to be a single cell line SK-N-BE2c for a bulk of the second half of the manuscript. Suggestions to for changes to the manuscript are highlighted below.

1. The authors demonstrate in Figure 1 that KDM6B expression is higher than for 6A or UTY. However, it has been published in Ref 62, that expression of KDM6B seems to be higher in lower stage/risk tumors which can be readily verified in the R2 database for the 649 tumor Kocak dataset. Furthermore, high expression of KDM6B in this dataset is associated with improved survival. A more in-depth discussion of this seems warranted as these data contradict those of the present manuscript.
2. It is striking that KDM6B inhibition appears to be effective in the SK-N-BE2c but not its parent line SK-N-BE2 (Figure 3F and Ref 41). While these lines do show some phenotypic differences in their ability to interconvert between N and S type, they are otherwise incredibly similar including MYCN amplification and KDM6B expression (Maris cell line dataset, R2 database). What happens to MYCN upon knockdown of KDM6B in SK-N-BE2?
3. Similarly, MYCN, E2F are known to be tightly linked in neuroblastoma cell lines and regulate cell cycle. Thus, it is not surprising that a compound like GSK-J4 would regulate these pathways in BE2c cells. However, it makes one wonder if this again a MYCN specific phenomenon. Given that the data in Figure 2 demonstrating activation of KDM6B in non-amplified lines, are these findings replicable in non-amplified neuroblastoma cell lines? Without those data, would it be possible that KDM6B depletion, either genetically or pharmacologically, is causing decreased growth via an alternative mechanism that reads out like a MYCN/E2F pathway in this one cell line?
4. Figure 3 appears to be addressing the questions above, however it is unclear which cell lines are in Group A vs Group B. Additionally, while the authors claim that E2F8 is important for distinguishing A from B, they then use a E2F signature to differentiate neuroblastoma signatures. Can the authors clarify why they switched and if E2F8 expression is indeed a biomarker in neuroblastoma cells?
5. The authors state that Palbociclib did not enhance the effect of GSK-J4, but that there was enhancement with JQ-1 or 17-DMAG. It is difficult to discern the difference between the plates presented in the supplement and those of the main manuscript. I would suggest a formal test of synergy to support the statement that there is enhancement with the two later drugs.
6. As the authors are aware, not all stage 4 patients are high-risk and not all high-risk patients are stage 4. Thus, while Figure 7 is certainly very intriguing, it may make more sense to include or replace the analysis by stage 4 (Figures 7E and 7G) with an inclusion of high-risk only as looking only within the subgroup of high-risk patients would likely be more clinically relevant as they would be expected to have more similar outcomes.
7. Are Hi-C data replicable in MYCN non-amplified neuroblastoma cell lines? The SKNAS data would suggest a very different TAD structure around E2F8 than was shown for SKNDZ.
8. Similar to above comments, it is somewhat unclear if the authors are focused only on the role of KDM8B on the CDK4/6-Rb-E2F pathway in MYCN amplified disease. Without more data to suggest a role in non-amplified disease, the authors may wish to consider amending their title to reflect the more limited role in MYCN amplified neuroblastoma.

Reviewer #3 (Remarks to the Author):

The study by Oto et al. "KDM6B promotes oncogenic CDK4/6-pRB-E2F pathway via maintaining enhancer activation in high-risk neuroblastoma" unravels a molecular mechanism controlled by the lysine demethylase KDM6B directing proliferation. The authors show that KDM6B is upregulated in neuroblastoma and neuroblastoma-derived cell lines. Genome-wide knockdown or inhibition of KDM6B reveals that it is involved in CDK4/6-pRB-E2F-directed cell cycle regulation. Consistently, a gene signature dependent on KDM6B is correlated with a reduced survival rate. Surprisingly, the occupation of H3K4me1 levels are repressed by KDM6B inhibition, while H3K27me3 is only mildly affected. Unfortunately, the authors fail to provide a molecular explanation for this interesting phenotype. The study, however, is very extensive and convincing.

Specific points:

- Please provide molecular explanation why the occupation of H3K4me1 levels are repressed by KDM6B inhibition, while H3K27me3 is only mildly affected.
- Fig.11: Please repeat the Western blot decorated with α -KDM6B (upper panel); differences in expression levels are not obvious.
- Please include size marks for all Western blots and scales for peak height for all images derived from genome-wide sequencing.
- Fig.S7F is not depicted.
- Page 10, 2nd paragraph, line 3: change to Figure 5A/B

Reviewer #4 (Remarks to the Author):

The manuscript describes the involvement of KDM6B in neuroblastoma oncogenesis through activation via disruption of the G1 checkpoint. Inhibition of KDM6B activity by GSK-J4 results in anti tumor activity by inhibiting this pathway.

The paper has major flaws in scientific reasoning and many results can be explained by off target effects.

Major comments are

1. Authors claim overexpression of KDM6B. In fact they only show expression of the mRNA in relation to a selected set of neural crest progenitors and in relation to other genes from the same class. This claim is not correct. First, comparing neuroblastoma to trunk neural crest cells is not correct. Various studies have shown that neuroblastoma develop from earlier progenitors so this is not a correct control population. If analyzed in relation to other tumor types and normal tissues there is no overexpression of KDM6B. Only Leukemias show overexpression of the gene which is in line with the tumor driving function in this type of malignancies (Data analysis performed by Reviewer in the R2 bioinformatic program that has been used by the authors as well). More extensive analysis of the expression of KDM6B in neuroblastoma datasets show that there is in fact low expression of KDM6B in poor prognosis tumors including MYCN amplified tumors. The gene is located on chromosome 17P which is not gained in high risk neuroblastoma but is sporadically deleted. These findings all do not support an oncogenic role for this gene in neuroblastoma. The finding presented by the authors seem to be selected for, based on fitting the hypothesis, instead of being objective observations.
2. Histone modification analysis are claimed to show epigenetic upregulation of KDM6B expression. The histone modifications only indicate that there is expression of the gene. Super enhancer activation, as is claimed, could be of interests but in this data that is not shown. The H3K27Ac should then be shown in relation to H3K27Ac in the rest of the genome.
3. The regulation of MYCN by KDM6C is claimed by knock down in one line. These data is very

weak since knock down of KDM6C is not appearing in the figure 1I top lane. This is also in contradiction with the clinical observation that there is no correlation between MYCN expression and KDM6B expression.

4. The results in figure 2 are completely based on the silencing of KDM6B in SKNBE. But as figure 1I shows there is hardly any knock down. Therefore this is most likely a result of off target effects resulting in a cell cycle arrest. Thereby the E2F signature will pop up.

5. The authors then continue with the compound GSK-J4 as a specific inhibitor of the KDM6B protein but as has been shown before this is a very aspecific inhibitor (<https://doi.org/10.1038/nature11262>). Results therefore might as well result from aspecific effects resulting in a cell cycle arrest and E2F signature. Correlations in NB cell lines can be based on cell growth characteristics where fast growing cell lines have higher E2F signatures and thereby are more sensitive for an a-specific inhibitor.

6. Authors claim in figure 6 that CDK4 and CDK6 confers resistance to CDK4/6 inhibitors. Instead; the cells with CDK-4 or 6 overexpression seem to grow faster and the relative growth inhibition does not seem to be changed. Colony forming assays are not the way to present this. This should be shown in relative dose response curves.

7. Finally the authors claim that a KDM6B signature defines certain risk groups in neuroblastoma. What we are looking at is a well known cluster of cell cycle and DNA damage pathway genes which is strongly correlated with prognosis in neuroblastoma. This gene signature apparently pops up when cells are exposed to an a-specific inhibitor (GDK-J4) that results in a cell cycle arrest.

Responses to reviewers:

We thank reviewers for their constructive comments, suggestions and critiques, which we believe have led to significant improvement of our manuscript.

Reviewers' comments:

Reviewer #2 (Remarks to the Author):

To investigate the role of KDM6B in regulating neuroblastoma. The authors present a wide swath of data to show that KDM6B is upregulated in neuroblastoma, has open and transcribed chromatin and that this gene plays a role in E2F and MYCN. The use both genetic and pharmacologic mechanisms of inhibition to demonstrate the robust impact of this gene and its inhibition. These data certainly do contribute to our understanding of the epigenetic regulation of neuroblastoma and the role of histone modifications. However, there are some limitations in this manuscript, particular the heavy reliance on what appears to be a single cell line SK-N-BE2c for a bulk of the second half of the manuscript. Suggestions to for changes to the manuscript are highlighted below.

We thank the reviewer's positivity for our manuscript and constructive suggestions. To overcome the limitation of our study that we heavily relied on single cell line SK-N-BE2c, we have performed the following experiments to address the reviewer's concerns:

KDM6B RNAi in KELLY, SK-N-DZ, SK-N-BE2, SK-N-FI cell lines, followed by western blot assessment of MYC expression, with results showing that MYCN or C-MYC were downregulated by KDM6B knockdown in all tested lines (**Supplementary Figure 3C-F**).

RNA-seq of additional cell lines including KELLY (MYCN amplified), SK-N-AS and SK-N-FI (both non-MYCN amplified) after KDM6B knockdown, followed by pathway analysis. The results showed that genetic knockdown of KDM6B significantly altered the gene signatures of Rb1-E2F in KELLY and SK-N-FI cells with less effect on SK-N-AS cells (**Supplementary Figure 4**).

We re-analyzed the published RNA-seq data for neuroblastoma cells treated with GSK-J4 (Sci Transl Med, 2018, 10(441):eaao4680). The results showed that Rb1-E2F pathways were also significantly downregulated by KDM6 inhibition (**Supplementary Figure 7**).

1. The authors demonstrate in Figure 1 that KDM6B expression is higher than for 6A or UTY. However, it has been published in Ref 62, that expression of KDM6B seems to be higher in lower stage/risk tumors which can be readily verified in the R2 database for the 649 tumor Kocak dataset. Furthermore, high expression of KDM6B in this dataset is associated with improved survival. A more in-depth discussion of this seems warranted as these data contradict those of the present manuscript.

We thank the reviewer for referring to Ref 62 so that we have the opportunity to clarify the difference between our study and that study. We agreed that the expression of KDM6B in high stage neuroblastoma is lower and higher levels of KDM6B seemed to be correlated with a better survival.

Currently, we do not understand why KDM6B expression is relatively lower in high stage disease although the KDM6B absolute value still remains very high in high-stage tumors. Considering KDM6B is one epigenetic modifier that could be plastic in different cellular contexts, very high levels of KDM6B might not be compatible with the environment of high stage cancer. While we commonly infer the potential importance of a gene by correlating their expression levels with disease stage or survival, it is not always straightforward to jump to a conclusion that the gene is not essential to cancer cells. Indeed, our data demonstrated that KDM6B is an essential gene in multiple neuroblastoma cell lines (both MYCN amplified and non-MYCN amplified) in which we used 4 different siKDM6B oligos to knockdown KDM6B (Cancer Research 2017, 77:4626-4638). In support of our data, neuroblastoma is the third most sensitive malignancy among 25 different cancer lineages that are sensitive to KDM6B knockdown (**Supplementary Figure 3A**). Importantly, we performed rescue experiments that showed KDM6B knockdown reduced MYCN expression and colony formation, which was rescued by RNAi resistant KDM6B cDNA (**Figure 1J-L**). Lastly, we would like to cite Dr. William Kaelin's opinion "Whether an elevated level of a particular protein is associated with a poor prognosis in a given cancer provides very little information as to whether that protein would be a good target in that cancer. Being associated with a poor prognosis is neither necessary nor sufficient to be a good cancer target" (Nature Reviews Cancer, 2017; 17:441–450). Our data demonstrate that KDM6B is a relevant target in neuroblastoma, even though the expression level of KDM6B may not be associated with poor prognosis in this disease.

Nevertheless, this reviewer raised a great and interesting question that warrants further study in the future to dissect why KDM6B has lower expression in high stage disease but still is essential to cancer cell survival. We have added sentences in the Discussion section by saying "Interestingly, the expression of KDM6B in high stage neuroblastoma is lower and higher levels of KDM6B seemed to be correlated with a better survival. Currently, we do not understand why KDM6B expression is relatively low in high stage disease (However, the KDM6B absolute value still remains very high in high-stage tumors). Considering KDM6B is one epigenetic modifier that could be plastic in different cellular contexts, very high levels of KDM6B might not be compatible to the environment of high stage cancer. While we commonly infer the potential importance of a gene by correlating their expression levels with disease stage or survival, it is not always straightforward to jump a conclusion that the gene is or not essential to cancer cells. Nevertheless, it raises an interesting question that warrants further study in the future to dissect why KDM6B lowers its expression in high stage disease but still is essential to cancer cell survival".

2. It is striking that KDM6B inhibition appears to be effective in the SK-N-BE2c but not its parent line SK-N-BE2 (Figure 3F and Ref 41). While these lines do show some phenotypic differences in their ability to interconvert between N and S type, they are otherwise incredibly similar including MYCN amplification and KDM6B expression (Maris cell line dataset, R2 database). What happens to MYCN upon knockdown of KDM6B in SK-N-BE2?

As the reviewer pointed out, SK-N-BE2c cells are a subclone of SK-N-BE2 and thus may have distinct features. We have knocked down KDM6B in SK-N-BE2 cells and assessed MYCN expression. It turned out that KDM6B depletion also reduced MYCN expression (**Supplementary Figure 3C**), indicating that KDM6B is also essential to SK-N-BE2. We cannot definitely explain why SK-N-BE2c is relatively more sensitive to GSK-J4 than SK-N-BE2. However, we noticed that SK-N-BE2 cells have a longer doubling time than SK-N-BE2c, and thus the lower sensitivity of SK-N-BE2 to KDM6B inhibition may be due to decreased cellular proliferation. However, in Figure 3B, among the different cancer cell lines sensitive to GSK-J4 treatment, SK-

N-BE2 is the only neuroblastoma cell line. Thus, this may suggest that neuroblastoma has unique genetic/epigenetic features that render it sensitive to GSK-J4, although SK-N-BE2 itself is less sensitive to GSK-J4 in comparison to other neuroblastoma cell lines.

3. Similarly, MYCN, E2F are known to be tightly linked in neuroblastoma cell lines and regulate cell cycle. Thus, it is not surprising that a compound like GSK-J4 would regulate these pathways in BE2c cells. However, it makes one wonder if this again a MYCN specific phenomenon. Given that the data in Figure 2 demonstrating activation of KDM6B in non-amplified lines (we believe reviewer meant MYCN amplified line), are these findings replicable in non-amplified neuroblastoma cell lines? Without those data, would it be possible that KDM6B depletion, either genetically or pharmacologically, is causing decreased growth via an alternative mechanism that reads out like a MYCN/E2F pathway in this one cell line?

We thank the reviewer for pointing out the alternative possibility.

We have knocked down KDM6B in 3 additional cell lines (KELLY with MYCN amplification, SK-N-AS and SK-N-FI without MYCN amplification) for RNA-seq analysis. Indeed, E2F pathways in MYCN amplified cell lines are more significantly affected (Supplementary Figure 4). In addition, we re-analyzed the published RNA-seq data (Sci Transl Med, 2018, 10(441):eaao4680) and found that indeed E2F pathways were significantly inhibited by GSK-J4 treatment in neuroblastoma cell lines. We have included these data in our revision (Supplementary Figure 7).

4. Figure 3 appears to be addressing the questions above, however it is unclear which cell lines are in Group A vs Group B. Additionally, while the authors claim that E2F8 is important for distinguishing A from B, they then use a E2F signature to differentiate neuroblastoma signatures. Can the authors clarify why they switched and if E2F8 expression is indeed a biomarker in neuroblastoma cells?

We have listed the cell line names (Supplementary Table 1).

While we believe that E2F8 is indeed a biomarker based on our chemogenetics data analysis, it is one of the transcripts of E2F transcriptome. We think E2F signature and E2F8 can be exchangeable in this case.

5. The authors state that Palbociclib did not enhance the effect of GSK-J4, but that there was enhancement with JQ-1 or 17-DMAG. It is difficult to discern the difference between the plates presented in the supplement and those of the main manuscript. I would suggest a formal test of synergy to support the statement that there is enhancement with the two later drugs.

We have performed new experiments, followed by analysis of the synergy using Bliss model. We have included the new data (Figure 6F, Supplementary Figure 14B-G).

6. As the authors are aware, not all stage 4 patients are high-risk and not all high-risk patients are stage 4. Thus, while Figure 7 is certainly very intriguing, it may make more sense to include or replace the analysis by stage 4 (Figures 7E and 7G) with an inclusion of high-risk only as looking only within the subgroup of high-risk patients would likely be more clinically relevant as they would be expected to have more similar outcomes.

We have replaced Figure 7E and 7G using high-risk group, and moved stage 4 data to Supplementary Figure 15.

7. Are Hi-C data replicable in MYCN non-amplified neuroblastoma cell lines? The SKNAS data would suggest a very different TAD structure around E2F8 than was shown for SKNDZ.

As the reviewer knows, TAD domains are usually stable, even across different species. We checked Hi-C data of BE2C, KELLY and SK-N-AS cells. E2F8 is localized in same TAD with NAV2 locus in all these cell lines, suggesting this TAD domain is conserved in neuroblastoma cells. We have included these data (**Supplementary Figure 11**).

8. Similar to above comments, it is somewhat unclear if the authors are focused only on the role of KDM8B on the CDK4/6-Rb-E2F pathway in MYCN amplified disease. Without more data to suggest a role in non-amplified disease, the authors may wish to consider amending their title to reflect the more limited role in MYCN amplified neuroblastoma.

We thank this reviewer for this suggestion.

While we have performed RNA-seq in more cell lines including KELLY (MYCN amplified), SK-N-AS and SK-N-FI (non-MYCN amplified), the results showed more consistency in MYCN amplified cells than the non-MYCN amplified cells (**Figure 2 and Supplementary Figure 4**).

We therefore changed title to “KDM6B promotes activation of the oncogenic CDK4/6-pRB-E2F pathway by maintaining enhancer activity in MYCN-amplified neuroblastoma”.

Reviewer #3 (Remarks to the Author):

The study by Oto et al. “KDM6B promotes oncogenic CDK4/6-pRB-E2F pathway via maintaining enhancer activation in high-risk neuroblastoma” unravels a molecular mechanism controlled by the lysine demethylase KDM6B directing proliferation. The authors show that KDM6B is upregulated in neuroblastoma and neuroblastoma-derived cell lines. Genome-wide knockdown or inhibition of KDM6B reveals that it is involved in CDK4/6-pRB-E2F-directed cell cycle regulation. Consistently, a gene signature dependent on KDM6B is correlated with a reduced survival rate. Surprisingly, the occupation of H3K4me1 levels are repressed by KDM6B inhibition, while H3K27me3 is only mildly affected. Unfortunately, the authors fail to provide a molecular explanation for this interesting phenotype. The study, however, is very extensive and convincing.

We appreciate the reviewer’s positivity about our manuscript.

Specific points:

- Please provide molecular explanation why the occupation of H3K4me1 levels are repressed by KDM6B inhibition, while H3K27me3 is only mildly affected.

The net peak numbers induced by GSK-J4 were probably counterbalanced by reduction of PRC2 complex, which are well-known E2F targets. While the reviewer raised a very interesting question, we have no definitive

answer, and it warrants further study. Previous studies showed that KDM6 members coordinate with H3K4 methyltransferases (KMT2) to regulate the levels of H3K4.

After analysis of our RNA-seq data, we found that KDM6B knockdown caused aberrant splicing of KMT2C and KMT2D in all cell lines, which may lead to changes in protein translation, mRNA stability or protein function. We did not include these data as it will take a long term effort to validate the biological consequences of these alterations.

However, we have added two sentences in the Discussion section by saying “The H3K4me1 is basically catalyzed by KMT2 family methyltransferases. Previous studies have shown that KMT2 members can form complexes with KDM6 members, suggesting a concerted mechanism for transcriptional activation in which cycles of H3K4 methylation by KMT2 are linked with the demethylation of H3K27.”

- Fig.1I: Please repeat the Western blot decorated with α -KDM6B (upper panel); differences in expression levels are not obvious.

We have repeated knockdown and performed western blot to replace Fig. 1I.

- Please include size marks for all Western blots and scales for peak height for all images derived from genome-wide sequencing.

We have included the size marks and peak height in all relevant figures.

- Fig.S7F is not depicted.

Fig. S7F is changed to Fig. S10F and depicted.

- Page 10, 2nd paragraph, line 3: change to Figure 5A/B

Sorry for the typo. We have changed it.

Reviewer #4 (Remarks to the Author):

The manuscript describes the involvement of KDM6B in neuroblastoma oncogenesis through activation via disruption of the G1 checkpoint. Inhibition of KDM6B activity by GSK-J4 results in anti tumor activity by inhibiting this pathway.

The paper has major flaws in scientific reasoning and many results can be explained by off target effects.

We thank the reviewer for the valuable critiques. To address this reviewer’s concerns, we have performed extensive experiments (which have also been detailed in each comment), and all results support our original conclusion.

We have performed a rescue experiment by introducing KDM6B cDNA in BE2C cells, followed by KDM6B knockdown using RNAi to target the 3’UTR of endogenous KDM6B. The results showed that KDM6B cDNA rescued cell survival and MYCN expression, which further support our conclusion that KDM6B is essential to neuroblastoma cell survival, and this is not an off-target effect (**Figure 1J-1L**)

We have generated a cell line stably overexpressing KDM6B in BE2C cells, which reduces global levels of H3K27me3, increases MYCN expression and leads to faster cell proliferation (**Supplementary Figure 3G, 3H**).

We re-analyzed an independent, genome-wide shRNA screen (20 different shRNAs per gene) dataset across nearly 400 cancer cell lines (Data are publicly accessible. DepMap.org). The results showed that KDM6B but not KDM6A is essential to neuroblastoma. Neuroblastoma ranks 3rd in sensitivity to KDM6B knockdown among all cancer types tested, just below myeloma and prostate cancer lineages. However, overall KDM6A is not essential to most cancer lineages including neuroblastoma cells (**Supplementary Figure 3A, 3B**).

We have performed KDM6B knockdown in additional multiple neuroblastoma cell lines and the results all showed that MYCN or C-MYC were downregulated by loss of KDM6B (**Supplementary Figure 3C-F**).

We have performed RNA-seq analysis in more cell lines after KDM6B knockdown. Pathway analysis showed that E2F pathways were downregulated (**Supplementary Figure 4**).

We used AlphaLISA approach to test the in vitro inhibitory activity of GSK-J1 and its prodrug (GSK-J4) against purified KDMs (KDM2A, KDM3A, KDM4A, KDM5A, KDM6A and KDM6B) (**Supplementary Figure 5A-C**). We assessed the major histone methylations after cells treated with GSK-J4 (**Supplementary Figure 5D**). Our results showed that GSK-J4 induced cellular effect by selectively targeting KDM6 but not other KDMs.

Major comments are

1. Authors claim overexpression of KDM6B. In fact, they only show expression of the mRNA in relation to a selected set of neural crest progenitors and in relation to other genes from the same class. This claim is not correct. First, comparing neuroblastoma to trunk neural crest cells is not correct. Various studies have shown that neuroblastoma develop from earlier progenitors so this is not a correct control population.

It is widely accepted that neuroblastoma is an embryonal tumor and the earliest cell of origin for neuroblastoma is thought to be a neural crest cell specified to the sympathoadrenal lineage that has not received or responded to cues that determine neuronal or chromaffin cell fate (Nature Reviews Cancer 2014, 14:277–289). Cancer genomic sequencing studies reveal that neuroblastoma bears mutations of genes involved in neural crest lineage commitment and epithelial-to-mesenchymal transition during the migration of neural crest cells, which further support the concept (see review <https://doi.org/10.3389/fnmol.2019.00009>). Recent RNA-seq and super enhancer definitions identified two interconvertible populations of neuroblastoma cells, adrenergic and neural crest like cells (Nature Genetics, 2017, 49: 1261–1266; 49:1408-1413). Additionally, recent single cell RNA-seq studies further indicate that neuroblastoma exhibits heterogeneous cells of origins (Cancer Cell, 2020; 38: 716-733; <https://www.biorxiv.org/content/10.1101/2020.05.15.097469v1.full>, <https://www.biorxiv.org/content/10.1101/2020.06.22.164301v4>, <https://www.biorxiv.org/content/10.1101/2020.05.04.077057v1>), all of which are derived from neural crest cells.

While neural crest cells are multipotent, the trunk neural crest cells are determined to differentiate to sympathetic neurons (Developmental Biology, 6th edition), and thus, it is highly likely that neuroblastoma cells are derived from trunk neural crest. As the developmental process of neural crest cell development and

migration is highly dynamic, it is technically challenging to accurately define and capture the progenitor cells of sympathetic neurons. Although mouse models generated through TH- or DBH-driven MYCN suggest that neuroblastoma is derived from sympathetic progenitors, which might be directly arise from the trunk neural crest. Dr. Kevin Freeman's laboratory used MYCN oncogene expression directly converted primary neural crest cells into neuroblastoma in a mouse model (Oncogene, 2017, 36: 5075-5082). Dr. Margareta Wilhelm's laboratory demonstrated that human neural crest cells derived from iPS cells are also able to form neuroblastoma (<https://openarchive.ki.se/xmlui/handle/10616/46615>).

Based on these published and unpublished studies, we rationalize that trunk neural crest cells are an appropriate control for neuroblastoma since there are no other normal tissues that could serve as more representative normal cell of origin for neuroblastoma.

If analyzed in relation to other tumor types and normal tissues there is no overexpression of KDM6B. Only Leukemias show overexpression of the gene which is in line with the tumor driving function in this type of malignancies (Data analysis performed by Reviewer in the R2 bioinformatic program that has been used by the authors as well).

We believe that it would be more scientifically sound to compare the tumor and its matched normal tissues, like comparing apples to apples rather than comparing apples to oranges. Numerous studies have shown that gene expression controlled by super-enhancers is lineage-specific (Richard Young, Cell, 2016). Our data indicate that KDM6B is regulated by a super-enhancer. When we checked any known oncogenes (such as MDM2 and CCNE1 as shown in Figures below) and tumor suppressors using R2, none of them show overexpression or underexpression across all lineages of normal tissues or tumor tissues. Cellular context is essential to define the functions of a gene. For example, EZH2 and NOTCH are well known to act as oncogenes or tumor suppressors in different tissue lineages (Nature Medicine, 2012, 18: 298–302; Nature Genetics, 2010, 42: 181–185; Journal Experimental Medicine, 2011, 208:1931–1935). It would be more reasonable to compare the changes of cancer cells to its cell of origin.

[FIGURE REDACTED]

[FIGURE REDACTED]

More extensive analysis of the expression of KDM6B in neuroblastoma datasets show that there is in fact low expression of KDM6B in poor prognosis tumors including MYCN amplified tumors. The gene is located on chromosome 17P which is not gained in high risk neuroblastoma but is sporadically deleted. These findings all do not support an oncogenic role for this gene in neuroblastoma. The finding presented by the authors seem to be selected for, based on fitting the hypothesis, instead of being objective observations.

Our data demonstrate that KDM6B is a relevant target in neuroblastoma, even though the expression level of KDM6B may not be associated with poor prognosis in this disease.

We would like to clarify that we did not claim KDM6B is an oncogene, rather we suggested it is essential to neuroblastoma cell survival and is involved in regulation of the E2F pathway. Despite this, oncogenic drivers have a broad definition. As this reviewer pointed out, KDM6B is located in 17p. While 17q21-qter as a minimal common region of gain, whole chromosome 17 gain is also seen in neuroblastoma reported two decades ago (Am J Pathol. 1997, 150: 81–89), and is widely accepted by the field. However, genetic abnormalities of 17p such as deletion were rarely seen in a pan-neuroblastoma genetic study of 702 cases recently reported from St Jude (Nature Communications, 2020; 11: 5183).

We agree that KDM6B is indeed expressed at relatively low levels in high stage or MYCN amplified tumors relative to other stages and non-MYCN amplified, whereas the absolute value of KDM6B is still very high, at least significantly higher than the well-known neuroblastoma oncogenes, LMO1 and LIN28B, based on the R2 dataset the reviewer used (see Figure below).

2. Histone modification analysis are claimed to show epigenetic upregulation of KDM6B expression. The histone modifications only indicate that there is expression of the gene. Super enhancer activation, as is claimed, could be of interests but in this data that is not shown. The H3K27Ac should then be shown in relation to H3K27Ac in the rest of the genome.

Our study was not only based on the histone modifications but we also showed absolute high levels of RNA expression of KDM6B based on RNA-seq data.

We do not entirely understand what “Super enhancer activation” is meant here but we infer this reviewer wanted us to compare the H3K27Ac signals of KDM6B and the rest of the genome by assuming the rest genomic regions show low or no H3K27Ac signals, as indicative of KDM6B super-enhancer activation.

The H3K27Ac in Figure 1G clearly showed that multiple clusters of H3K27Ac marks and other transcriptionally active marks occupy the 25kb KDM6B locus, which fits the definition of super enhancer (Cell, 2013. 153:307-319). While we did not show the histone marks across the whole genome (which could bear hundreds or thousands super-enhancers), we compared KDM6B with KDM6A and UTY, which showed very low signals of H3K27Ac, supporting that high expression of KDM6B is driven by a super-enhancer.

3. The regulation of MYCN by KDM6C is claimed by knock down in one line. These data is very weak since knock down of KDM6C is not appearing in the figure 1I top lane. This in also in contradiction with the clinical observation that there is no correlation between MYCN expression and KDM6B expression.

We believe this reviewer meant KDM6B but not KDM6C.

We have performed a rescue experiment by introducing KDM6B cDNA in BE2C cells, followed by KDM6B knockdown using RNAi to target the 3’UTR of endogenous KDM6B. The results showed that KDM6B cDNA rescued cell survival and MYCN expression (Figure 1J-1L), which further supports our conclusion that KDM6B is essential for neuroblastoma cell survival, and this is not an off-target effect. We have performed KDM6B knockdown in multiple cell lines and the results all showed that MYCN or C-MYC were downregulated by loss of KDM6B. We have generated cell lines stably overexpressing KDM6B in BE2C cells, which reduces global levels of H3K27me3, increases MYCN expression and leads to faster cell proliferation (Supplementary Figure 3G, 3H). We re-analyzed an independent, genome-wide shRNA screen (20 different shRNAs per gene) dataset across nearly 400 cancer cell lines (Data are publicly accessible. DepMap.org). The results showed that KDM6B but not KDM6A is essential to neuroblastoma. Neuroblastoma ranks 3rd in

sensitivity to KDM6B knockdown, just below myeloma and prostate cancer lineages. However, overall KDM6A is not essential to most cancer lineages including neuroblastoma cells (**Supplementary Figure 3A, 3B**). We have performed KDM6B knockdown in additional multiple neuroblastoma cell lines and the results all showed that MYCN or C-MYC were downregulated by loss of KDM6B (**Supplementary Figure 3C-F**). Taken together, all these data support our conclusion.

We have no knowledge of clinical studies for correlation between MYCN and KDM6B being reported. While we agree that, after checking gene expression database, no correlation of *MYCN* and *KDM6B* expression was found, it does not mean KDM6B is not essential and won't regulate MYCN expression. Our data clearly demonstrate that KDM6B regulates MYCN expression. In support of this, a recent study also showed KDM6B regulates MYCN expression in pluripotent stem cells, namely, overexpression of KDM6B increases MYCN expression while knockdown of KDM6B reduces MYCN expression (Nat Commun. 2020;11(1):5061. doi: 10.1038/s41467-020-18900-z).

4. The results in figure 2 are completely based on the silencing of KDM6B in SKNBE. But as figure 1I shows there is hardly any knock down. Therefore, this is most likely a result of off target effects resulting in a cell cycle arrest. Thereby the E2F signature will pop up.

Please see our response above. Our data support an on-target rather than an off-target effect.

In addition, we don't believe that an off-target effect of knockdown of a gene will invariably lead to cell cycle arrest and thereby E2F signature will pop up.

5. The authors then continue with the compound GSK-J4 as a specific inhibitor of the KDM6B protein but as has been shown before this is a very aspecific inhibitor (<https://doi.org/10.1038/nature11262>). Results therefore might as well result from aspecific effects resulting in a cell cycle arrest and E2F signature. Correlations in NB cell lines can be based on cell growth characteristics where fast growing cell lines have higher E2F signatures and thereby are more sensitive for an a-specific inhibitor.

Here we would like to look back at the history of the dispute about GSK-J4. GSK company developed a selective and potent KDM6 inhibitor, GSK-J1, (Nature, 2012. 488: 404–408), which is unfortunately unable to penetrate cells. To make a cell-permeable inhibitor, GSK-J1 was modified by adding an ester (named as GSK-J4) specifically to enable efficient intracellular delivery of the compound. Once it enters cells, GSK-J4 converts to GSK-J1 and then is able to inhibit KDM6. Thus, GSK-J4 is not itself a chemical tool for direct KDM inhibition, but was designed specifically to enable efficient intracellular delivery of GSK-J1 into cells. The intracellular conversion of ester pro-drug is complete within 15 min after which levels of intracellular GSK-J4 are negligible (Nature, 2014, 514:E2).

Nevertheless, as this reviewer pointed out, a following study by the Helin group showed that GSK-J1 could hit KDM5 at higher concentrations (Nature, 2014, 514:E1). However, their data was strongly refuted by the GSK group in a published rebuttal (Nature, 2014, 514:E2). We also questioned the Helin group's data for the following reasons. (1) The Helin group used U2OS osteosarcoma cells to assess the H3K27me2 levels, which is an intermediate between H3K27me3 and H3K27me1, and can therefore represent either active methylation or demethylation of H3K27 residues. Thus, it is neither an ideal marker to assess the activity of KDM6-mediated H3K27 demethylation in cells, nor in an *in vitro* biochemical assay. (2) Most importantly, our group published a paper showing that U2OS cells do not express any H3K27me3 or H3K27me2 due to the

defects of PRC2 complex (Oncotarget. 2018. 9: 27087–27091). Therefore, the H3K27 methyl signals from that study may be due to non-specific signal from the antibody that they used.

To validate and profile the selectivity of GSK-J1/GSK-J4, we used AlphaLISA approach, a method that tests the *in vitro* inhibiting activity of GSK-J1 and GSK-J4 against purified KDMs (KDM2A, KDM3A, KDM4A, KDM5A, KDM6A and KDM6B), the main histone lysine demethylase families. We confirmed that GSK-J1 was selectively active against KDM6A and KDM6B, had some effect on KDM5A but showed no activity to KDM2A, KDM3A and KDM4A (**Supplementary Figure 5B**), and this result is consistent with the published GSK data. Consistent with its mechanism of action, GSK-J4 showed no activity against all tested KDMs (**Supplementary Figure 5C**). While GSK-J4 could have off-target effects when administered at high concentrations, just as any compound, the concentration we used in cells (2.5 μ M) only induced the upregulation of H3K27me3, but not H3K9me3, H3K36me3 and H3K4me3 (**Supplementary Figure 5D**), three other major histone methylations of H3, indicating that GSK-J4 induced cellular effect by selectively targeting KDM6 but not other KDMs. Additionally, a selective KDM5 inhibitor, KDM5-C70, showed no effect on neuroblastoma cell proliferation even administered at very high concentrations (40 μ M) (**Supplementary Figure 5E, 5F**)

6. Authors claim in figure 6 that CDK4 and CDK6 confers resistance to CDK4/6 inhibitors. Instead; the cells with CDK-4 or 6 overexpression seem to grow faster and the relative growth inhibition does not seem to be changed. Colony forming assays are not the way to present this. This should be shown in relative dose response curves.

We thank the reviewer for the suggestion. We have performed dose response curve and the results showed that indeed, overexpression of CDK4/6 confers resistance to GSK-J4 and palbociclib (**Figure 6I, J**).

7. Finally the authors claim that a KDM6B signature defines certain risk groups in neuroblastoma. What we are looking at is a well known cluster of cell cycle and DNA damage pathway genes which is strongly correlated with prognosis in neuroblastoma. This gene signature apparently pops up when cells are exposed to an a-specific inhibitor (GDK-J4) that results in a cell cycle arrest.

As we discussed above in response to comment 5 of this Reviewer, GSK-J4 is selectively against KDM6 in cells at the concentrations we used. The cell cycle and DNA damage pathway genes induced by GSK-J4 actually further supports that this inhibitor works through KDM6B, whose knockdown induced similar phenotype.

REVIEWER COMMENTS

Reviewer #2 (Remarks to the Author):

The authors have been very responsive to critique and have provided sufficient additional supportive data and editing. One thing I find very curious is the comment about the connection to KDM6B and EZH2 and they presumably have competing and opposite effects on H3K27. Can the authors comment more about that?

Reviewer #3 (Remarks to the Author):

Remarks to the authors

The study by Oto et al. "KDM6B promotes oncogenic CDK4/6-pRB-E2F pathway via maintaining enhancer activation in high-risk neuroblastoma" unravels a molecular mechanism controlled by the lysine demethylase KDM6B directing proliferation. The authors show that KDM6B is upregulated in neuroblastoma and neuroblastoma-derived cell lines. Genome-wide knockdown or inhibition of KDM6B reveals that it is involved in CDK4/6-pRB-E2F-directed cell cycle regulation. Consistently, a gene signature dependent on KDM6B is correlated with a reduced survival rate. Surprisingly, the occupation of H3K4me1 levels are repressed by KDM6B inhibition, while H3K27me3 is only mildly affected. Unfortunately, the authors fail to provide a molecular explanation for this interesting phenotype. The study, however, is very extensive and convincing.

Specific points:

- Please provide molecular explanation why the occupation of H3K4me1 levels are repressed by KDM6B inhibition, while H3K27me3 is only mildly affected.
- Fig.11: Please repeat the Western blot decorated with α -KDM6B (upper panel); differences in expression levels are not obvious.
- Please include size marks for all Western blots and scales for peak height for all images derived from genome-wide sequencing.
- Fig.S7F is not depicted.
- Page 10, 2nd paragraph, line 3: change to Figure 5A/B

Reviewer #5, replacement reviewer for Reviewer #4(Remarks to the Author):

The authors have addressed those questions raised by the reviewers. The authors have shown the similarities between genetic silencing of KDM6B and pharmacologic inhibition of KDM6B at regulating colony formation and gene transcription. It will be nice to show whether the knockdown of KDM6B have a similar effect on regulating epigenetic changes compared to the GSK-J4 treatment.

Responses to reviewers

We thank the reviewers for their constructive comments, suggestions and critiques, which we believe have led to significant improvement of our manuscript after its substantial revision. We have addressed all of the reviewers' comments and critiques in a point-by-point manner, which were highlighted in blue color.

Reviewer #2 (Remarks to the Author):

The authors have been very responsive to critique and have provided sufficient additional supportive data and editing. One thing I find very curious is the comment about the connection to KDM6B and EZH2 and they presumably have competing and opposite effects on H3K27. Can the authors comment more about that?

We thank this reviewer for his/her positive comments.

The connection of KDM6B and EZH2 is indeed very interesting. Since EZH2 is the downstream target of E2F and MYCN, we assumed the reduction of EZH2 by KDM6B inhibition is through this mechanism. However, considering both proteins are epigenetic modifiers that confer cellular plasticity, we also speculate that cells strive for adaptation to the KDM6B inhibition by suppressing EZH2 in order to counterbalance the net increase of H3K27me3. We have added this discussion on page17, lines 514-518.

Reviewer #3 (Remarks to the Author):

Remarks to the authors

The study by Oto et al. "KDM6B promotes oncogenic CDK4/6-pRB-E2F pathway via maintaining enhancer activation in high-risk neuroblastoma" unravels a molecular mechanism controlled by the lysine demethylase KDM6B directing proliferation. The authors show that KDM6B is upregulated in neuroblastoma and neuroblastoma-derived cell lines. Genome-wide knockdown or inhibition of KDM6B reveals that it is involved in CDK4/6-pRB-E2F-directed cell cycle regulation. Consistently, a gene signature dependent on KDM6B is correlated with a reduced survival rate. Surprisingly, the occupation of H3K4me1 levels are repressed by KDM6B inhibition, while H3K27me3 is only mildly affected. Unfortunately, the authors fail to provide a molecular explanation for this interesting phenotype. The study, however, is very extensive and convincing.

We really thank this reviewer for the insightful and critical comment. Since reviewer 5 also wondered if KDM6B knockdown and GSK-J4 treatment would show similarities, we applied a new approach, CUT&Tag, and repeated the whole experiment to assess H3K27me3 and H3K4me1 in response to KDM6B knockdown and GSK-J4 treatment (Figure 5 and Supplemental Figure 11). CUT&Tag overcomes the shortcomings of ChIP-seq and has exceptionally low background. In addition, the residual DNA from *E. Coli* in CUT&Tag reagents serves as a spike-in for internal control, making the bioinformatics analysis more accurate.

The patterns of H3K27me3 and H3K4me1 changes were indeed similar between KDM6B knockdown and GSK-J4 treatment, indicating that the observed phenotype reflects the mechanism of action of KDM6B inhibition. Particularly, we analyzed the transcriptional factor motif enrichment for altered H3K27me3 and H3K4me1 peaks. We found that KDM6B inhibition specifically leads to increased H3K27me3 at CTCF and

BORIS (also named CTCFL) binding sites. However, H3K4me1 at CTCF and BORIS binding sites is selectively reduced. These data suggest that increased H3K27me3 displaces H3K4me1 modifiers (highly likely the H3K4me1 methyltransferase KMT2) from CTCF and BORIS sites, consequently disrupting the enhancer activity. Previous studies have shown that enhancers, CTCF and H3K4me1 peaks overlap. CTCF regulates the long-range chromatin interactions at enhancers. Thus, KDM6B inhibition may disrupt the chromatin interaction at its target genes such as MYCN and E2F targets. Indeed, our CUT&Tag data showed that the super-enhancer region of MYCN, which exhibits long-range chromatin interactions within a TAD domain, has the most significant changes of H3K27me3 and H3K4me1 by KDM6B knockdown and GSK-J4 treatment, in line with the reduction of MYCN expression by KDM6B inhibition.

We have replaced the old data with new data (p10-11, lines 306-353), and given more discussions (p16, lines 484-497).

Additionally, we would like to call your attention to the H3K4me1 data. While the H3K27me3 peaks were modestly increased by KDM6B inhibition, the H3K4me1 also seemed to have more peaks called out in KDM6B knockdown and GSK-J4 treatment groups. However, the final conclusion drawn from both experiments remains the same for interpretation of mechanism of KDM6B inhibition. We thought there might be several reasons causing the discrepancies between our CUT&Tag and previous ChIP-seq. First, it could be due to technical reasons since the experimental procedures between ChIP-seq and CUT&Tag are different. Second, we normalized ChIP-seq H3K4me1 by using a computational algorithm while the CUT&Tag H3K4me1 was normalized by E. coli DNA spike-in as an internal control, which is more accurate. We believe our CUT&Tag data may represent the bona fide epigenetic changes in cells. Therefore, we used CUT&Tag to replace the ChIP-seq data in Figure 5. Nevertheless, the final conclusion stands. We appreciate this reviewer for giving us the opportunity to revise our methods and approaches to draw a more solid conclusion.

Specific points:

- Please provide molecular explanation why the occupation of H3K4me1 levels are repressed by KDM6B inhibition, while H3K27me3 is only mildly affected.

Please see response above.

- Fig.1I: Please repeat the Western blot decorated with α -KDM6B (upper panel); differences in expression levels are not obvious.

We have repeated the western blot and replaced the old data with a new version.

- Please include size marks for all Western blots and scales for peak height for all images derived from genome-wide sequencing.

We have added the size marks and scales for peak height.

- Fig.S7F is not depicted.

We have depicted it.

- Page 10, 2nd paragraph, line 3: change to Figure 5A/B

We have replaced it with new content.

Reviewer #5, replacement reviewer for Reviewer #4(Remarks to the Author):

The authors have addressed those questions raised by the reviewers. The authors have shown the similarities between genetic silencing of KDM6B and pharmacologic inhibition of KDM6B at regulating colony formation and gene transcription. It will be nice to show whether the knockdown of KDM6B have a similar effect on regulating epigenetic changes compared to the GSK-J4 treatment.

We thank this reviewer for the critical comment. To test if KDM6B knockdown and GSK-J4 induce similar epigenetic phenotypes, we applied a new approach, CUT&Tag, and repeated the whole experiment to assess H3K27me3 and H3K4me1 in response to KDM6B knockdown and GSK-J4 treatment (Figure 5 and Supplemental Figure 11). Our results indeed showed that KDM6B and GSK-J4 induced similar epigenetic changes, particularly for the KDM6B targets such as *MYCN* oncogene. Please see details above in response to reviewer 3. We have replaced the old data with new data (p10-11, lines 306-353), and given more discussions (p16, lines 484-497).

REVIEWERS' COMMENTS

Reviewer #2 (Remarks to the Author):

The authors have adequately addressed all of my concerns.

Reviewer #5 (Remarks to the Author):

The questions have been well-addressed.